# Impact of energy limitations on function and resilience in long-wavelength Photosystem II

**Stefania Viola[1]\*, William Roseby[1], Stefano Santabarbara[2], Dennis Nürnberg[3], Ricardo Assunção[3], Holger Dau[3], Julien Sellés[4], Alain Boussac[5], Andrea Fantuzzi[1], A William Rutherford[1]\***

[1]Department of Life Sciences, Imperial College London, London, United Kingdom; [2]Photosyntesis Research Unit, Consiglio Nazionale delle Ricerche, Milan, Italy; [3]Physics Department, Freie Universität Berlin, Berlin, Germany; [4]Institut de Biologie Physico-Chimique, UMR CNRS 7141 and Sorbonne Université, Paris, France; [5]Institut de Biologie Intégrative de la Cellule, UMR9198, CEA Saclay, Gif-Sur-Yvette, France

**\*For correspondence:**
s.viola@imperial.ac.uk (SV);
a.rutherford@imperial.ac.uk (AWR)

**Competing interest:** The authors declare that no competing interests exist.

**Abstract** Photosystem II (PSII) uses the energy from red light to split water and reduce quinone, an energy-demanding process based on chlorophyll a (Chl-a) photochemistry. Two types of cyano-bacterial PSII can use chlorophyll d (Chl-d) and chlorophyll f (Chl-f) to perform the same reactions using lower energy, far-red light. PSII from *Acaryochloris marina* has Chl-d replacing all but one of its 35 Chl-a, while PSII from *Chroococcidiopsis thermalis*, a facultative far-red species, has just 4 Chl-f and 1 Chl-d and 30 Chl-a. From bioenergetic considerations, the far-red PSII were predicted to lose photochemical efficiency and/or resilience to photodamage. Here, we compare enzyme turnover efficiency, forward electron transfer, back-reactions and photodamage in Chl-f-PSII, Chl-d-PSII, and Chl-a-PSII. We show that: (i) all types of PSII have a comparable efficiency in enzyme turnover; (ii) the modified energy gaps on the acceptor side of Chl-d-PSII favour recombination via $P_{D1}^+Phe^-$ repopulation, leading to increased singlet oxygen production and greater sensitivity to high-light damage compared to Chl-a-PSII and Chl-f-PSII; (iii) the acceptor-side energy gaps in Chl-f-PSII are tuned to avoid harmful back reactions, favouring resilience to photodamage over efficiency of light usage. The results are explained by the differences in the redox tuning of the electron transfer cofactors Phe and $Q_A$ and in the number and layout of the chlorophylls that share the excitation energy with the primary electron donor. PSII has adapted to lower energy in two distinct ways, each appropriate for its specific environment but with different functional penalties.

## Editor's evaluation

This manuscript describes the energetic mechanisms by which two quite different cyanobacteria use far-red light. The work describes the energetic constraints and preferred operating conditions of these "strategies" in particular on how nature has solved the problem of low energy "headroom'" required to prevent deleterious back reactions while maintaining efficient energy storage. The differences between the species are quite interesting and show that nature has evolved multiple solutions to fundamental limitations. Given the importance of understanding and improving the efficiency of photosynthesis, and the new insights revealed, the work will be of interest to a broad audience.

**eLife digest** Algae, plants and cyanobacteria perform a process called photosynthesis, in which carbon dioxide and water are converted into oxygen and energy-rich carbon compounds. The first step of this process involves an enzyme called photosystem II, which uses light energy to extract electrons from water to help capture the carbon dioxide.

If the photosystem absorbs too much light, compounds known as reactive oxygen species are produced in quantities that damage the photosystem and kill the cell. To ensure that the photosystem works efficiently and to protect it from damage, about half of the energy from the absorbed light is dissipated as heat, while the rest of the energy is stored in the products of photosynthesis.

The standard form of photosystem II uses the energy of visible light, but some cyanobacteria contain different types of photosystem II, which do the same chemical reactions using lower energy far-red light. One type of far-red photosystem II is found in *Acaryochloris marina*, a cyanobacterium living in stable levels of far-red light, shaded from visible light. The other type is found in a cyanobacterium called *Chroococcidiopsis thermalis,* which can switch between using its far-red photosystem II when shaded from visible light and using its standard photosystem II when exposed to it. Being able to work with less energy, the two types of far-red photosystem II appear to be more efficient than the standard one, but it has been unclear if there were any downsides to this trait.

Viola et al. compared the standard photosystem II with the far-red photosystem II types from *C. thermalis* and *A. marina* by measuring the efficiency of these enzymes, the quantity of reactive oxygen species produced, and the resulting light-induced damage. The experiments revealed that the far-red photosystem II of *A. marina* is highly efficient but produces elevated levels of reactive oxygen species if exposed to high light conditions. On the other hand, the far-red photosystem II of *C. thermalis* is less efficient in collecting and using far-red light, but is more robust, producing fewer reactive oxygen species.

Despite these tradeoffs, engineering crop plants or algae that could use far-red photosynthesis may help boost food and biomass production. A better understanding of the trade-offs between efficiency and resilience in the two types of far-red photosystem II could determine which features would be beneficial, and under what conditions. This work also improves our knowledge of how the standard photosystem II balances light absorption and damage limitation to work efficiently in a variable environment.

## Introduction

Photosystem II (PSII) is the water/plastoquinone photo-oxidoreductase, the key energy converting enzyme in oxygenic photosynthesis. The near-universal type of PSII, found in all photosynthetic eukaryotes and in most cyanobacteria, contains 35 chlorophylls a (Chl-a) and 2 pheophytins a (Phe). Four of the Chl molecules ($P_{D1}$, $P_{D2}$, $Chl_{D1}$, and $Chl_{D2}$) and both Phe molecules are located in the reaction centre (***Diner and Rappaport, 2002***). The remaining 31 Chl-a in the PSII core constitute a peripheral light-collecting antenna. When antenna chlorophylls are excited by absorbing a photon, they transfer the excitation energy to the primary electron donor, $Chl_{D1}$, the red-most chlorophyll in the reaction centre, although it's been reported that charge separation from $P_{D1}$ can occur in a fraction of centres (***Diner and Rappaport, 2002***; ***Holzwarth et al., 2006***; ***Romero et al., 2010***; ***Cardona et al., 2012***). The initial charge separation, forming the first radical pair $Chl_{D1}^+Phe^-$ (assuming $Chl_{D1}$ as primary donor), is quickly stabilized by the formation of the second radical pair, $P_{D1}^+Phe^-$, and then by further electron transfer steps (***Figure 1A***) that lead to the reduction of plastoquinone and the oxidation of water.

PSII activity is energy demanding. In Chl-a-PSII, the primary donor absorbs red photons at 680 nm, and this defines the energy available for photochemistry (1.82 eV) with a high quantum yield for the forward reactions. The energy stored in the products of the reaction (reduced plastoquinone and molecular oxygen) and in the trans-membrane electrochemical gradient is ~1 eV, while the remaining ~0.82 eV is released as heat helping to ensure a high quantum yield for the forward reaction and minimize damaging and wasteful side- and back-reactions. The 1.82 eV was suggested to be the minimum amount of energy required for an optimum balance of efficiency versus resilience to photodamage, and responsible for explaining the 'red limit' (~680 nm) for oxygenic photosynthesis (***Nürnberg et al., 2018***; ***Rutherford et al., 2012***).

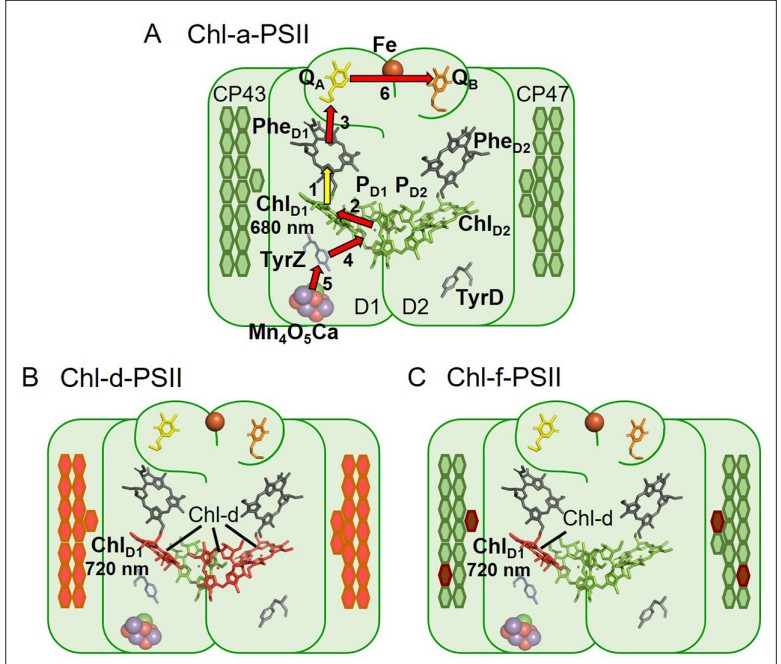

**Figure 1.** The three types of PSII. (**A**) Chl-a-PSII with the key cofactors of the reaction centre, located in the subunits D1 and D2, labelled. Besides the $P_{D1}$, $P_{D2}$, $Chl_{D1}$, and $Chl_{D2}$ chlorophylls and the two pheophytins, $Phe_{D1}$ and $Phe_{D2}$, these cofactors include the quinones, $Q_A$ and $Q_B$, and the non-heme iron (Fe) on the acceptor side and the two redox-active tyrosines TyrZ and TyrD and the manganese cluster ($Mn_4O_5Ca$) on the donor side. The arrows represent the electron transfer steps and the numbers the order of the steps. The yellow arrow is the primary charge separation, with other steps shown as red arrows. The primary donor is shown as $Chl_{D1}$. (**B**) and (**C**) Chl-d-PSII and Chl-f-PSII, with the far-red chlorophylls in the reaction centres highlighted and the wavelength of the primary donor, assumed to be $Chl_{D1}$, indicated. The hexagons on the sides of each reaction centre represent the chlorophylls of the respective antennas, located in the subunits CP43 and CP47. Chl-a is represented in green, Chl-d in orange and Chl-f in brown. In (**C**) the single Chl-d is located in the $Chl_{D1}$ position, reflecting the assignment of the single Chl-d as the primary donor (*Gisriel et al., 2022*), leaving the remaining 4 Chl-f molecules as peripheral antenna. For all three types of PSII, the model of the reaction centre cofactors was made based on the crystal structure of PSII from the cyanobacterium *Thermosynechococcus vulcanus* (PDB ID: 3WU2, *Umena et al., 2011*).

The first reported case in which the red limit is exceeded was the chlorophyll d (Chl-d)-containing cyanobacterium *Acaryochloris marina* (*A. marina*) (*Miyashita et al., 1996*). Chl-d-PSII contains 34 Chl-d and 1 Chl-a (proposed to be in the $P_{D1}$ position *Renger and Schlodder, 2008*) and uses less energy, with the proposed Chl-d primary donor in the $Chl_{D1}$ position absorbing far-red photons at ~720 nm (*Schlodder et al., 2007*), corresponding to an energy of ~1.72 eV (*Figure 1B*).

Recently, it was discovered that certain cyanobacteria use an even more red-shifted pigment, chlorophyll f (Chl-f), in combination with Chl-a (*Chen et al., 2010*; *Gan et al., 2014*). When grown in far-red light, these cyanobacteria replace their Chl-a-PSII with Chl-f-PSII, that has far-red specific variants of the core protein subunits (D1, D2, CP43, CP47, and PsbH) and contains ~90% of Chl-a and ~10% of Chl-f (*Nürnberg et al., 2018*; *Gan et al., 2014*). The Chl-f-PSII from *Chroococcidiopsis thermalis* PCC7203 (*C. thermalis*), which contains 30 Chl-a, 4 Chl-f, and 1 Chl-d, was shown to have a long wavelength primary donor (originally proposed to be either Chl-f or d, in the $Chl_{D1}$ position *Nürnberg et al., 2018*) absorbing far-red photons at ~720 nm (*Figure 1C*), the same wavelength as in *A. marina* (*Nürnberg et al., 2018*; *Judd et al., 2020*). A recent cryo-EM structure has also argued for $Chl_{D1}$ being the single Chl-d in the Chl-f-PSII of *Synechococcus* sp. PCC7335 (*Gisriel et al., 2022*). This suggests that this could be the case also in the Chl-f-PSII of *C. thermalis*, because of the conservation of the amino acids coordinating $Chl_{D1}$ in the far-red PSII of the two species. The facultative, long-wavelength species that use Chl-f are thus the second case of oxygenic photosynthesis functioning

beyond the red-limit (**Nürnberg et al., 2018**), but the layout of their long wavelength pigments is quite different from that of the Chl-d-PSII.

Assuming that Chl-a-PSII already functions at an energy red limit (**Rutherford et al., 2012**), the diminished energy in Chl-d-PSII and Chl-f-PSII seems likely to increase the energetic constraints. Thus, if the far-red PSII variants store the same amount of energy in their products and electrochemical gradient, as seems likely, then it was suggested that they should have decreased photochemical efficiency and/or a loss of resilience to photodamage (**Nürnberg et al., 2018**; **Cotton et al., 2015**; **Davis et al., 2016**). These predicted energetic constraints are worth investigating to generate knowledge that could be beneficial for designing strategies aimed at engineering of far-red photosynthesis into other organisms of agricultural or technological interest (**Chen and Blankenship, 2011**).

Here we report a comparison of the enzyme turnover efficiency, forward reactions, and back-reactions in the three known types of PSII: Chl-a-PSII, and the two far-red types, the Chl-f-PSII from *C. thermalis* and the Chl-d-PSII from *A. marina*. To compare the enzymatic properties of the three types of PSII and minimize the effects of physiological differences between strains, isolated membranes rather than intact cells were used. The use of isolated membranes allows the minimization of potential effects due to: (i) the transmembrane electric field, which affects forward electron transfer (**Diner and Joliot, 1976**) and charge recombination (**Joliot and Joliot, 1980**), (ii) the uncontrolled redox state of the plastoquinone pool in whole cells, which can affect the $Q_B/Q_B^-$ ratio present in dark-adapted PSII, (iii) differences in the size and composition of the phycobilisomes and in their association with PSII, and (iv) the presence of photoprotective mechanisms such as excitation energy quenching and scavengers of reactive oxygen species.

## Results

### Fluorescence decay kinetics in the three types of PSII

The electron transfer properties of the three types of PSII were investigated by comparing the decay kinetics of the flash-induced fluorescence in membranes from *A. marina*, white-light (WL) grown *C. thermalis* and far-red-light (FR) grown *C. thermalis*. When forward electron transfer occurs (**Figure 2A**), the fluorescence decay comprises three phases (**Crofts and Wraight, 1983**; **Vass et al., 1999**): the fast phase (~0.5ms) is attributed to electron transfer from $Q_A^-$ to $Q_B$ or $Q_B^-$ and the middle phase (~3ms) is generally attributed to $Q_A^-$ oxidation limited by plastoquinone (PQ) entry to an initially empty

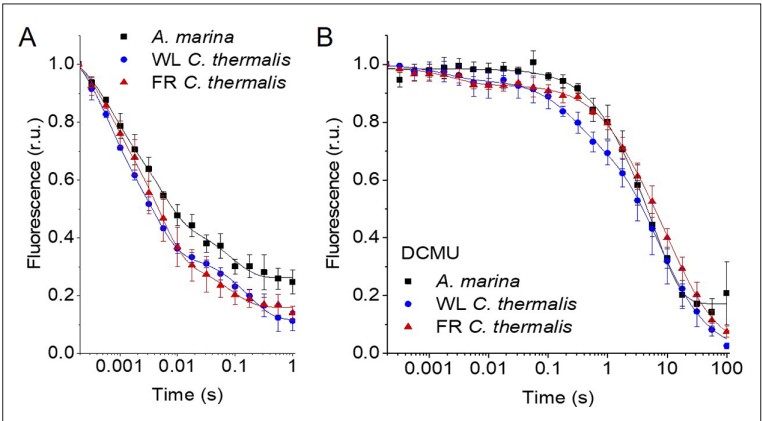

**Figure 2.** Fluorescence decay kinetics after a saturating flash in membranes of *A. marina*, WL *C. thermalis* and FR *C. thermalis* with no additions (**A**) and in presence of DCMU (**B**). The datapoints represent the averages of three biological replicates,± s.d. (provided in **Figure 2—source data 1**), the lines represent the fits of the experimental data. All traces are normalized on the initial variable fluorescence ($F_m$-$F_0$, with $F_m$ measured 190 μs after the saturating flash). The full 100 s traces of the data in (**A**) are shown in **Figure 2—figure supplement 1**.

The online version of this article includes the following source data and figure supplement(s) for figure 2:

**Source data 1.** Fluorescence decay kinetics.

**Figure supplement 1.** Fluorescence decay kinetics after a short saturating light pulse in isolated membranes of *A. marina*, WL *C. thermalis* and FR *C. thermalis*.

**Table 1.** Time constants and relative amplitudes (%) of the different phases of fluorescence decay obtained by fitting the data in *Figure 2* and *Figure 2—figure supplement 1*.

Statistically significant differences according to Student's t-tests are indicated with asterisks (*p≤0.05, **p≤0.01, ***p≤0.001, ****p≤0.0001).

**No addition (1 s)[*]**

| | Fast phase | Middle phase | Slow phase +$y_0$ |
|---|---|---|---|
| **Strain** | **T1/Amp (ms/%)** | **T2/Amp (ms/%)** | **Amp (%)** |
| *A. marina* | 0.58±0.21 / 26±5 | 4.9±1.3 / 32±5 | 42±3** |
| *C. thermalis* WL | 0.50±0.09 / 32±3 | 3.7±0.4 / 37±4 | 31±2 |
| *C. thermalis* FR | 0.53±0.16 / 26±4 | 4.7±0.7 / 45±4 | 30±3 |

**No addition 100 s[†]**

| | Not bound | Middle phase | Slow phase |
|---|---|---|---|
| **Strain** | **T1/Amp (ms/%)** | **T2/Amp (s/%)** | **T3/Amp (s/%)** |
| *A. marina* | -/- | 0.98±0.58 / 19±8 | 6.5±1.0 / 81±8 |
| *C. thermalis* WL | 2.0±0.9 / 5±1 | 0.25±0.04 / 17±1 | 6.9±0.3 / 78±1 |
| *C. thermalis* FR | 2.7±0.9 / 6±1 | 1.31±0.35** / 14±3 | 10.4±0.8** / 80±3 |

**DCMU (100 s)[‡]**

| | Fast phase | Middle phase | Slow phase |
|---|---|---|---|
| **Strain** | **T1/Amp (ms/%)** | **T2/Amp (ms/%)** | **T3/Amp (s/%)** |
| *A. marina* | 1.8±0.3 / 47±3*** | 44.7±11.2 / 26±3 | 10.8±2.6* / 27±1**** |
| *C. thermalis* WL | 1.7±0.2 / 62±2 | 99.8±23.5*/ 24±2 | 5.6±2.4 / 14±2 |
| *C. thermalis* FR | 2.2±0.3 / 58±3 | 38.7±10.3 / 26±3 | 14.3±4.6* / 16±1 |

[*]The decay kinetics measured over 100 s in samples with no additions were truncated at 1 s and fitted with a three exponential equation allowing $y_0$ to account for the part decaying in >1 s. For this reason, the cumulative amplitude of the slowest exponential decay phase and of $y_0$ is provided, but no $T_3$.

[†]The decay kinetics recorded over a period of 100 s were fitted with two exponentials and one hyperbole. In the case of *A. marina*, fitting of the fluorescence decay kinetics was done by excluding the datapoints between 30 and 100 s after flash, because of the presence of a non-decaying fluorescence that likely arises from a fraction of centres devoid of an intact Mn-cluster in which $Q_A^-$ is stabilised.

[‡]The data recorded in the presence of DCMU over a period of 100 s were fitted with two exponentials (only one in the case of *A. marina*) and one hyperbole.

$Q_B$ site and/or by $Q_BH_2$ exiting the site prior to PQ entry (*de Wijn and van Gorkom, 2001*). These two phases had comparable time-constants in all samples ($T_1$=0.5–0.6 and $T_2$=3.5–5ms, *Table 1*). The fast electron transfer from $Q_A^-$ to the non-heme iron possibly oxidized in a fraction of centres is too fast (t½~50 µs) to be detected here.

The slower decay phase is attributed to the charge recombination between $Q_A^-$ and the Mn-cluster mostly in the $S_2$ state (see section 2.2) in centres where forward electron transfer to $Q_B/Q_B^-$ did not occur. This phase was significantly slower in FR *C. thermalis* ($T_3$=14.3 ± 4.6 s) than in WL *C. thermalis* ($T_3$=5.6 ± 2.4 s) but had a similar amplitude in the two samples (*Figure 2—figure supplement 1* and *Table 1*). In *A. marina* this phase had a bigger amplitude than in the two *C. thermalis* samples (*Table 1*), because it was superimposed to a non-decaying component of the fluorescence, that did not return to the original $F_0$ level even at 100 s after the flash (*Figure 2—figure supplement 1*). This non-decaying component, absent in the two *C. thermalis* samples, is attributed to centres without a functional Mn-cluster, in which $P_{D1}^+$ is reduced by an electron donor that does not recombine in the minutes timescale (such as $Mn^{2+}$, TyrD, or the ChlZ/Car side-path), with the consequence of stabilizing $Q_A^-$ (*Nixon et al., 1992*; *Debus et al., 2000*). The fluorescence decay arising from the $S_2Q_A^-$ recombination was slower in *A. marina* ($T_3$=10.8 ± 2.6 s) than in WL *C. thermalis*, but its overlap with the non-decaying component made the fit of its time-constant potentially less reliable.

Indeed, when the fluorescence decay due to charge recombination was measured in presence of the $Q_B$-site inhibitor DCMU (*Figure 2B*), the decay kinetics were bi-phasic in all samples, and no difference in the major $S_2Q_A^-$ recombination phase (slow phase in *Table 1*, ~80% amplitude, $T_3$ ~6–7 s) was

found between *A. marina* and WL *C. thermalis*. In contrast, the decay was significantly slower in FR *C. thermalis*, with the time-constant of the major $S_2Q_A^-$ recombination phase (slow phase in *Table 1*, ~80% amplitude, $T_3=10.4 \pm 0.8$ s) similar to that measured in the absence of DCMU. The shorter lifetime (~0.22–1 s) of the middle decay phase (amplitude 15–20%) was compatible with it originating from TyrZ•(H+)$Q_A^-$ recombination occurring either in centres lacking an intact Mn-cluster (*Yerkes et al., 1983*) or in intact centres before charge separation is fully stabilised, as proposed in *Debus et al., 2000*. The fluorescence decay in WL and FR *C. thermalis* also had an additional fast phase of small amplitude (5–6%), attributed to forward electron transfer in centres in which DCMU was not bound (*Lavergne, 1983*). Again, the *A. marina* traces included a non-decaying phase of fluorescence, attributed to centres lacking an intact Mn-cluster.

The fluorescence decay kinetics in membranes of *Synechocystis* sp. PCC6803 (*Synechocystis*), perhaps the best studied Chl-a containing cyanobacterium, were also measured as an additional control. The kinetics in *Synechocystis* membranes were comparable to those reported for WL *C. thermalis* (Appendix 1). The *Synechocystis* and *A. marina* fluorescence decay kinetics measured in membranes here are overall slower than those previously measured in cells (*Cser et al., 2008*). This difference is ascribed to pH and membrane potential effects, as discussed in Appendix 1, and illustrates the difficulty to use whole cells for such measurements.

To conclude, the forward electron transfer rates from $Q_A^-$ to $Q_B/Q_B^-$ are not significantly different in the three types of PSII. In contrast, the $S_2Q_A^-$ recombination is slower in Chl-f-PSII of FR *C. thermalis* compared to Chl-a-PSII of WL *C. thermalis* and Chl-d-PSII of *A. marina*.

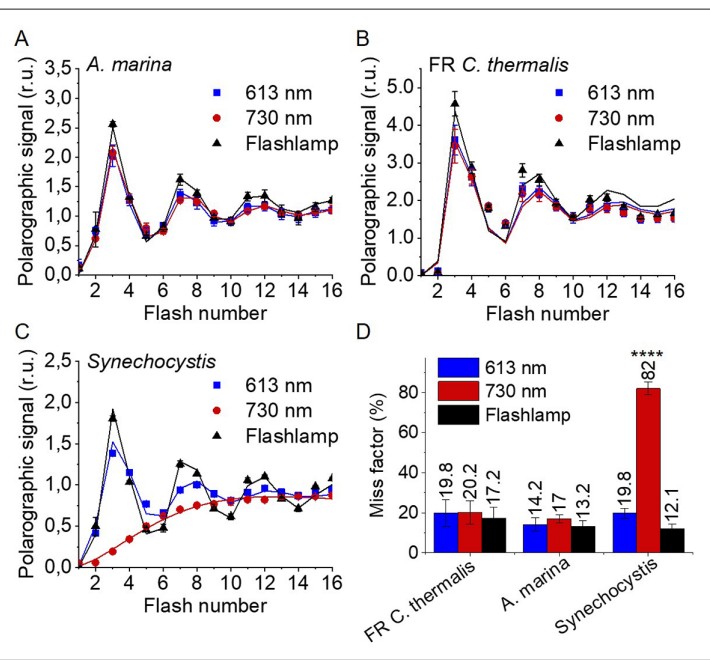

**Figure 3.** Flash-induced release of O₂ measured by polarography. (**A–C**) Patterns of oxygen release in *A. marina*, FR *C. thermalis* and *Synechocystis* membranes. Flashes were given at 900 ms intervals and the O₂ produced after each flash was measured. Flashes were provided by a white xenon flash lamp, a red LED centered at 613 nm, and a far-red LED centered at 730 nm. The data represent the averages of 3 biological replicates ±s.d. The lines represent the fits of the experimental data. The data were normalized to the O₂ yield of the last of the 40 flashes sequence. The non-normalized data are shown in *Figure 3—figure supplement 1*. Normalized and non-normalized data are provided in *Figure 3—source data 1*. (**D**) Miss factors (in %) calculated from the data shown in (**A–C**). The miss factor in *Synechocystis* membranes flashed at 730 nm is significantly higher than in *A. marina* and FR *C. thermalis* membranes according to Student's t-test, as indicated with asterisks (****p≤0.0001).

The online version of this article includes the following source data and figure supplement(s) for figure 3:

**Source data 1.** Flash-dependent oxygen evolution.

**Figure supplement 1.** Flash-induced release of O₂ measured by polarography.

**Table 2.** Initial distribution of S-states obtained by fitting the flash-dependent oxygen evolution data in *Figure 3*.

| | FR *C. thermalis* | | | *A. marina* | | | *Synechocystis* | | |
|---|---|---|---|---|---|---|---|---|---|
| | 613 nm LED | 730 nm LED | Flashlamp | 613 nm LED | 730 nm LED | Flashlamp | 613 nm LED | 730 nm LED | Flashlamp |
| $S_0$ (%) | 15.7 | 15.3 | 14.9 | 15.6 | 15.9 | 15.2 | 22.3 | 18.7 | 19.7 |
| $S_1$ (%) | 84.3 | 84.7 | 85.1 | 75.4 | 76.1 | 75.8 | 66.7 | 73.3 | 71.3 |
| $S_2$ (%) | 0 | 0 | 0 | 9 | 8 | 9 | 11 | 8 | 9 |

## S-state turnover efficiency in the far-red PSII

The efficiency of PSII water oxidation activity can be estimated by the flash-dependent progression through the S-states of the Mn-cluster. This can be measured by thermoluminescence (TL), which arises from radiative recombination of the $S_2Q_B^-$ and $S_3Q_B^-$ states (*Rutherford et al., 1982*). The TL measured in *A. marina*, WL *C. thermalis,* and FR *C. thermalis* membranes showed similar flash-dependencies in all three types of PSII (*Appendix 2—figure 1*), confirming and extending the earlier report (*Nürnberg et al., 2018*). Because the TL data presented some variability between biological replicates (Appendix 2), additional analyses were performed by polarography and absorption spectroscopy.

*Figure 3* shows the flash-dependent oxygen evolution measured in *A. marina*, FR *C. thermalis* and *Synechocystis* membranes. The latter were used as a Chl-a-PSII control because the content of PSII in membranes of WL *C. thermalis* was too low to allow accurate $O_2$ polarography measurements (*Figure 3—figure supplement 1D*). As shown by fluorescence, no significant difference in forward electron transfer between the two types of Chl-a-PSII was observed (Appendix 1), and the use of *Synechocystis* membranes was therefore considered as a valid control.

The measurements were performed using white, red, and far-red flashes. As expected, in dark-adapted samples, with $S_1$ as the majority state (*Table 2*), the maximal $O_2$ evolution occurred on the 3rd flash with subsequent maxima at 4 flash intervals. These maxima reflect the occurrence of the $S_3Y_Z^\bullet/S_4$ to $S_0$ transition in most centres as two water molecules are oxidized, resulting in the release of $O_2$. This oscillation pattern was the same in all samples and under all excitation conditions, except in *Synechocystis* membranes illuminated with far-red light, where the slow rise in $O_2$ evolution is due to the weak excitation of Chl-a-PSII by the short wavelength tail of the 730 nm flash.

The miss factor, indicating the fraction of PSII centres failing to progress through the S-states after a saturating flash excitation (*Lavergne, 1991*; *Grabolle and Dau, 2007* and see Discussion ), was ≤20% in all the samples except in the *Synechocystis* sample illuminated with far-red flashes, where it was >80% (*Figure 3D*). For *A. marina,* the misses (13–17%) were very similar to those reported earlier (*Shevela et al., 2006*). The misses in FR *C. thermalis* and in *Synechocystis* when illuminated with the 613 nm LED were slightly higher (17%–20%). Nevertheless, these differences, attributed to the combination of the absence of exogenous electron acceptors, and the relatively long and possibly not fully saturating flashes (*Figure 3—figure supplement 1*), were not significant.

In order to confirm and expand the results obtained with polarography, we measured the S-state turnover as the flash-induced absorption changes at 291 nm (*Figure 4*), that reflect the redox state of the Mn ions in the oxygen evolving complex (*Lavergne, 1991*; *Boussac et al., 2004*). These measurements were done in the presence of the electron acceptor PPBQ and using single-turnover monochromatic saturating laser flashes. In the case of *A. marina*, the measurements could be done using membranes, but the membranes of WL and FR *C. thermalis* could not be used because of their high light-scattering properties in the UV part of the spectrum. In the case of the FR *C. thermalis* partially purified $O_2$ evolving Chl-f-PSII were made and used for the measurements, while difficulties were encountered in isolating $O_2$ evolving PSII from WL *C. thermalis*. Therefore, PSII cores from *T. elongatus* with the D1 isoform PsbA3 (*Sugiura et al., 2008*) were used as a Chl-a-PSII control. Among the three D1 present in *T. elongatus*, PsbA3 has the highest sequence identity with the D1 of Chl-f-PSII in FR *C. thermalis* (see Discussion).

The Chl-f-PSII was illuminated with flashes at wavelengths preferentially absorbed by Chl-a (680 nm) and by long-wavelength chlorophylls (720–750 nm) (*Figure 4A*). As expected, maximum absorption decrease (positive $\Delta I/I$, as defined in Materials and methods, section UV transient absorption) occurred on $S_2$ (flash 1,5,9 etc.) and maximum absorption increase (negative $\Delta I/I$) on $S_0$ (flash 3,7,11 etc.) (*Lavergne, 1991*). No differences could be observed in either the amplitude or the damping of the

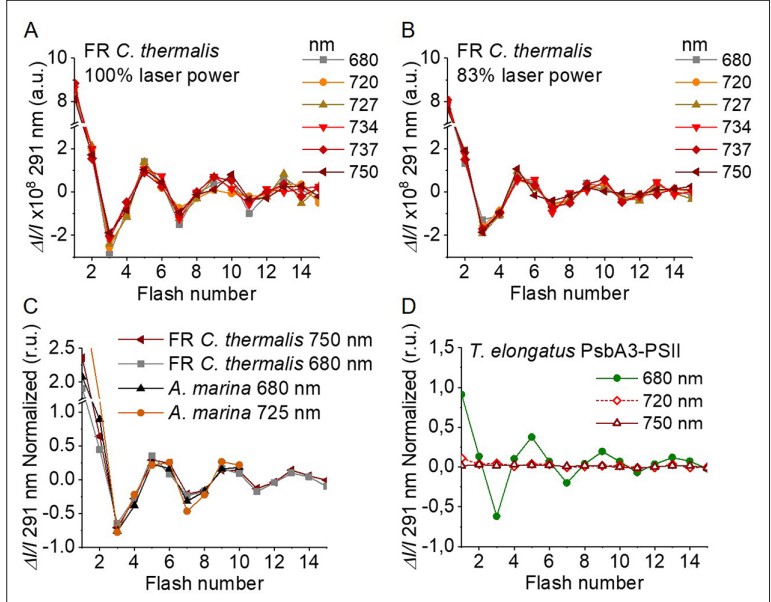

**Figure 4.** Flash-induced S-state turnover in FR *C. thermalis* PSII cores, *A. marina* membranes, and *T. elongatus* PsbA3-PSII cores. Absorption changes were measured at 291 nm at 100 ms after each of a series of single-turnover saturating flashes fired with a 300 ms time interval. (**A**) and (**B**) Measurements in FR *C. thermalis* PSII cores using flashes at the indicated wavelengths with 100% and 83% laser power (the power of the laser at the different wavelengths is reported in Appendix 7). (**C**) Comparison between the absorption changes obtained in FR *C. thermalis* PSII cores and *A. marina* membranes using flashes at the indicated wavelengths (100% laser power). The traces in (**C**) were normalized on the maximal oscillation amplitude (3rd minus 5th flash). The breaks in the vertical axes in panels (**A–C**) allow the oscillation pattern to be re-scaled for clarity, because the absorption change on the first flash contains a large non-oscillating component (*Lavergne, 1991*) that was not included in the fits. (**D**) Measurements in isolated *T. elongatus* PsbA3-PSII cores using flashes at the indicated wavelengths. All data are provided in *Figure 4—source data 1*.

The online version of this article includes the following source data for figure 4:

**Source data 1.** Flash-dependent UV absorption.

oscillations between the excitation wavelengths. When using sub-saturating flashes (~83% power), the damping of the oscillations was the same for all excitation wavelengths (*Figure 4B*), verifying that the illumination with 100% laser power was saturating at all the wavelengths. The equal amplitude of the oscillations obtained at all excitation wavelengths also indicates that the FR *C. thermalis* sample used does not contain any detectable Chl-a-PSII contamination. No differences in the oscillation patterns measured in FR *C. thermalis* Chl-f-PSII cores and in *A. marina* membranes, flashed at either 680 or 725 nm, were observed (*Figure 4C*). The PSII of *T. elongatus* showed a normal S-states progression when using 680 nm excitation, but no oscillation pattern when far-red flashes were used (*Figure 4D*). For all samples the calculated miss factor was ~10% (Appendix 3, discussion based on *Styring and Rutherford, 1987*; *Velthuys and Visser, 1975*; *Vermaas et al., 1984*; *Sugiura et al., 2004*).

In conclusion, the data reported here show that the overall efficiency of electron transfer from water to the PQ pool is comparable in all three types of PSII (independently of the Chl-a-PSII control used), as shown by the near-identical flash patterns of thermoluminescence (Appendix 2) and O$_2$ release (*Figure 3*), both measured without external electron acceptors. When the S-state turnover was measured by following the absorption of the Mn-cluster in the UV (*Figure 4*), the use of artificial electron acceptors and single-turnover saturating flashes allowed us to obtain better resolved flash patterns that were essentially indistinguishable in all three types of PSII and between excitation with visible or far-red light in the case of the Chl-d-PSII and Chl-f-PSII.

## Back-reactions measured by (thermo)luminescence

Charge recombination reactions were investigated by monitoring the thermoluminescence and luminescence emissions. The TL curves in *Figure 5A and B* show that both Chl-f-PSII and Chl-d-PSII are

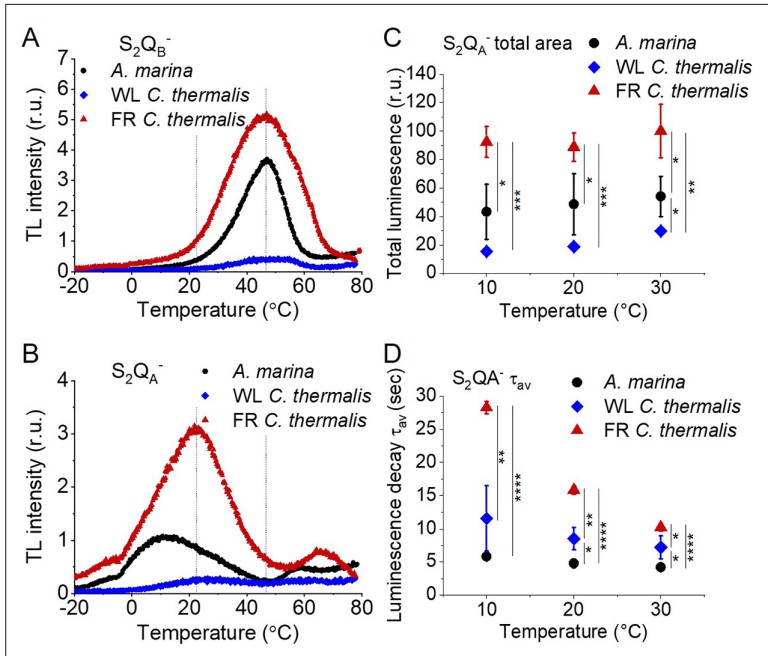

**Figure 5.** Thermoluminescence and luminescence measured in *A. marina*, WL *C. thermalis* and FR *C. thermalis* membranes. (**A**) and (**B**) TL measured in the absence of inhibitors ($S_2Q_B^-$) or in the presence of DCMU ($S_2Q_A^-$), respectively. The signal intensities are normalized on the content of $O_2$-evolving PSII of each sample, measured as the maximal oxygen evolution rates under saturating illumination. The dashed vertical lines indicate the two peak positions of the *C. thermalis* samples. (**C**) Plots of the total $S_2Q_A^-$ luminescence emission (integrated area below the curves), normalized on the maximal oxygen evolution rate of each sample, at 10, 20, and 30°C. (**D**) Plots of the average $S_2Q_A^-$ luminescence decay lifetimes ($\tau_{av}$), calculated from the decay phases attributed to $S_2Q_A^-$ recombination, as a function of temperature. In (**C**) and (**D**) each point represents the average of 3 biological replicates ±s.d. Statistically significant differences according to Student's t-tests are indicated with asterisks (*p≤0.05, **p≤0.01, ***p≤0.001, ****p≤0.0001).

The online version of this article includes the following source data and figure supplement(s) for figure 5:

**Figure supplement 1.** Thermoluminescence in *A. marina*, WL *C. thermalis* and FR *C. thermalis* membranes.

**Figure supplement 1—source data 1.** Thermoluminescence curves.

**Figure supplement 2.** Thermoluminescence and luminescence in *Synechocystis* membranes.

**Figure supplement 2—source data 1.** *Synechocystis* (thermo)luminescence curves.

more luminescent than Chl-a-PSII, with Chl-f-PSII being the most luminescent. These differences, that are much larger than the variability between biological replicates (*Figure 5—figure supplement 1C and D* and *Table 3*), fit qualitatively with earlier reports (*Nürnberg et al., 2018*; *Cser et al., 2008*) (see Appendix 4 for more details). The high luminescence indicates that in the Chl-d-PSII and Chl-f-PSII

**Table 3.** Average values (± s.d.) of the temperatures (T) and of the normalized amplitudes (Amp, in relative units) of the thermoluminescence peaks from $S_2Q_B^-$ and from $S_2Q_A^-$ back-reactions, plotted in *Figure 5—figure supplement 1*.

The difference in temperature between the $S_2Q_B^-$ and the $S_2Q_A^-$ (ΔT) is also reported. The ΔT in *A. marina* is significantly bigger than the one in WL and FR *C. thermalis* according to Student's t-test, as indicated with an asterisk (*p≤0.05).

| | $S_2Q_B^-$ | | $S_2Q_A^-$ | | |
|---|---|---|---|---|---|
| Strain | T (°C) | Amp (r.u.) | T (°C) | Amp (r.u.) | ΔT (°C) |
| *A. marina* | 46.5±1.8 | 2.77±1.15 | 14.9±3.7 | 1.15±0.55 | 31.5±2.8* |
| WL *C. thermalis* | 52.9±3 | 0.65±0.31 | 28.1±1.7 | 0.54±0.21 | 24.9±3.2 |
| FR *C. thermalis* | 50.3±4.7 | 4.61±1.29 | 26±3.3 | 3.27±0.88 | 24.3±5.1 |

there is an increase in radiative recombination, although the causes of this increase are likely to be different between the two photosystems, as detailed in the Discussion.

Despite the large difference in TL intensity between the Chl-a-PSII and Chl-f-PSII, the peak temperatures corresponding to the $S_2Q_B^-$ and $S_2Q_A^-$ recombination were both similar in Chl-a-PSII and Chl-f-PSII. In Chl-d-PSII, the temperature of the $S_2Q_B^-$ peak was only slightly lower, while the $S_2Q_A^-$ peak was ~15 °C lower (*Figure 5—figure supplement 1A and B* and *Table 3*). Earlier TL reports comparing Chl-d-PSII in *A marina* cells with Chl-a-PSII in *Synechocystis* cells also showed that, while the peak position of $S_2Q_B^-$ recombination was similar in the two samples, the $S_2Q_A^-$ peak position was lower in *A. marina* (*Cser et al., 2008*), in agreement with the present results in membranes. The peak temperatures measured in cells were lower than those reported here, which can be explained by (i) the effect of the transmembrane electric field, as discussed for the fluorescence decay results, and (ii) by differences in the heating rates used (1 °C s$^{-1}$ here, 0.33 °C s$^{-1}$ in *Cser et al., 2008*). When performing the same measurements in *Synechocystis* membranes (*Figure 5—figure supplement 2A*), the $S_2Q_B^-$ and $S_2Q_A^-$ peak positions were comparable to those obtained in the two *C. thermalis* samples, confirming that the lower $S_2Q_A^-$ peak temperature is a specific feature of Chl-d-PSII.

The $S_2Q_A^-$ recombination in the presence of DCMU was also measured by luminescence decay kinetics at 10, 20, and 30°C, a range of temperatures that covers those of the $S_2Q_A^-$ TL peaks of the three samples. Luminescence decay kinetics were recorded from 570ms for 300 seconds after the flash. In this time-range, the luminescence arises mainly from recombination *via* the back-reaction of $S_2Q_A^-$ (*Goltsev et al., 2009*). The total $S_2Q_A^-$ luminescence emission (*Figure 5C*) reflected the intensities of the TL peaks, as expected (*Rutherford and Inoue, 1984*), with the order of intensity as follows: Chl-f-PSII>Chl-d-PSII>Chl-a-PSII (although the variability between replicates made the difference between Chl-a-PSII and Chl-d-PSII less significant than that measured by TL). The total emissions did not vary significantly between 10°C and 30°C, although the decay kinetics were temperature-sensitive (*Appendix 5—figure 1*). The decay components identified by fitting the curves and their significance are discussed further in Appendix 5, based on *Yerkes et al., 1983*; *Lavorel and Dennery, 1984*; *Tyystjarvi and Vass, 2007*; *Sugiura et al., 2014*. The luminescence decay attributed to $S_2Q_A^-$ recombination was bi-phasic (*Appendix 5—table 1*), with the kinetics of both phases being faster in Chl-d-PSII (~3 and ~11 s) than in Chl-a-PSII (~4 and ~25 s), but slower in Chl-f-PSII (~9 and ~39 s). The average $S_2Q_A^-$ luminescence decay lifetimes accelerated with increasing temperature in Chl-a-PSII and Chl-f-PSII but were always the fastest in Chl-d-PSII and the slowest in Chl-f-PSII (*Figure 5D*). The luminescence decay kinetics of the Chl-a-PSII in *Synechocystis* membranes were similar to those measured in WL *C. thermalis* (*Figure 5—figure supplement 2B and C*), suggesting, as seen with the TL data, that the differences in kinetics observed in the two types of far-red PSII are not due to differences between species.

In conclusion, both Chl-f-PSII and Chl-d-PSII show strongly enhanced luminescence, as previously reported (*Nürnberg et al., 2018*; *Cser and Vass, 2007*). However, the Chl-d-PSII differs from the Chl-a-PSII and Chl-f-PSII by having a lower $S_2Q_A^-$ TL peak temperature and a faster $S_2Q_A^-$ luminescence decay. This indicates that Chl-d-PSII has a smaller energy gap between $Q_A^-$ and Phe compared to Chl-a-PSII and Chl-f-PSII, resulting in: (i) less heat required for the electron to be transferred energetically uphill from $Q_A^-$ to Phe (manifest as lower TL peak temperature), and (ii) a bigger proportion of $S_2Q_A^-$ recombination occurring via repopulation of $P_{D1}^+Phe^-$, a route faster than direct $P_{D1}^+Q_A^-$ recombination (manifest as faster luminescence decay kinetics, see also Appendix 5). The lower TL temperature and faster luminescence decay for $S_2Q_A^-$ recombination in Chl-d-PSII, but without a marked increase in its $Q_A^-$ decay rate as monitored by fluorescence (*Figure 2*), could reflect differences in the competition between radiative and non-radiative recombination pathways in Chl-d-PSII compared to those in Chl-a-PSII and Chl-f-PSII. In contrast, in Chl-f-PSII the energy gap between $Q_A^-$ and Phe does not appear to be greatly affected or could even be larger, as suggested by the slower $S_2Q_A^-$ recombination measured by fluorescence (*Figure 2*) and luminescence (*Figure 3*) decay. The $Q_B$ potentials appear to be largely unchanged, as manifested by the similar $S_2Q_B^-$ stability in all three types of PSII, with the slightly lower $S_2Q_B^-$ TL peak temperature in *A. marina* probably reflecting the decrease in the energy gap between $Q_A^-$ and Phe.

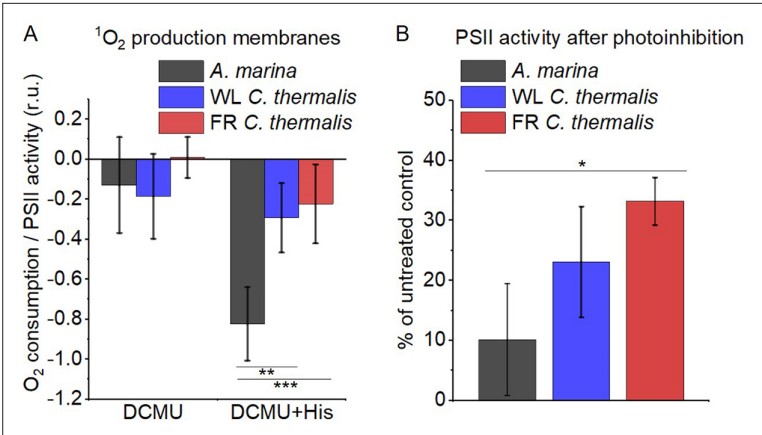

**Figure 6.** $^1O_2$ production and PSII sensitivity to high light in *A. marina*, WL *C. thermalis* and FR *C. thermalis* membranes. All samples were used at a chlorophyll concentration of 5 µg ml⁻¹. (**A**) $^1O_2$ production in presence of DCMU measured as the rate of histidine-dependent consumption of $O_2$ induced by saturating illumination (xenon lamp, 7100 µmol photons m⁻² s⁻¹, saturation curves in Appendix 6). The data are averages (± s.d.) of six biological replicates for *A. marina* and FR *C. thermalis* and three replicates for WL *C. thermalis* for the DCMU +His samples, and of four biological replicates for FR *C. thermalis* and three replicates for *A. marina* and WL *C. thermalis* for the DCMU samples. For each replicate, the rates of oxygen consumption were normalized to the maximal oxygen evolution rates measured in presence of DCBQ and ferricyanide. The non-normalized rates of each replicate are provided in Appendix 6. All traces are provided in *Figure 6—source data 1*. (**B**) Maximal PSII activities, measured as in (**A**), after 30 min illumination with saturating red light (660 nm LED, 2600 µmol photons m⁻² s⁻¹) relative to the maximal activities measured in control samples kept in darkness (provided in *Figure 6—source data 2*). The light used for the 30 min treatment was as saturating as the xenon lamp used in (**A**) (see Appendix 6). The data are averages of three biological replicates ±s.d. Statistically significant differences according to Student's t-tests are indicated with asterisks (*p≤0.05, **p≤0.01, ***p≤0.001).

The online version of this article includes the following source data and figure supplement(s) for figure 6:

**Source data 1.** Singlet oxygen production membranes.

**Source data 2.** Photoinhibition membranes.

**Figure supplement 1.** Singlet oxygen production in *A. marina*, WL *C. thermalis*, FR *C. thermalis* and *Synechocystis* membranes.

**Figure supplement 1—source data 1.** Singlet oxygen *Synechocystis*, *A. marina* sodium azide, Rose Bengal.

## Singlet oxygen production and sensitivity to high light in the far-red PSII

The smaller energy gap between $Q_A^-$ and Phe reported here in *A marina* is expected to result in enhanced singlet $O_2$ production and hence greater sensitivity to photodamage (*Nürnberg et al., 2018*; *Cotton et al., 2015*; *Johnson et al., 1995*; *Vass and Cser, 2009*). This was investigated by measuring the rates of $^1O_2$ generation induced by saturating illumination in isolated membranes using histidine as a chemical trap (*Figure 6A*, representative traces in *Figure 6—figure supplement 1A-C*). $^1O_2$ reacts with histidine to form the final oxygenated product, $HisO_2$, resulting in the consumption of $O_2$, as measured using the $O_2$ electrode. Without the histidine trap, most $^1O_2$ is thought to be quenched by carotenoids (*Telfer et al., 1994*). When histidine was present in addition to DCMU, the Chl-d-PSII in *A. marina* membranes showed significant light-induced $^1O_2$ formation. Under the same conditions, little $^1O_2$ formation occurred in Chl-a-PSII or Chl-f-PSII in *C. thermalis* membranes. Similarly low levels of $^1O_2$ were generated by Chl-a-PSII in *Synechocystis* membranes (*Figure 6—figure supplement 1D*). The His-dependent $O_2$ consumption in *A. marina* membranes showed the same light intensity dependence as $O_2$ evolution (*Appendix 6—figure 1B*), which suggests that $^1O_2$ formation was related to Chl-d-PSII photochemistry. Sodium azide, a $^1O_2$ quencher, suppressed the His-dependent oxygen consumption measured in *A. marina* in the presence of DCMU and when using the $^1O_2$-generating dye Rose Bengal, confirming that it was due to the production of $^1O_2$ (*Figure 6—figure supplement 1E and F*).

The strikingly high amount of $^1O_2$ generated by Chl-d-PSII prompted us to perform additional controls. (i) To test if the high $^1O_2$ production was related to the intactness of the PSII donor side, Mn was removed from *A. marina* membranes by Tris-washing. This had little effect on the $^1O_2$ formation with respect to the Mn-containing membranes (*Appendix 6—figure 2*), suggesting that the high $^1O_2$ production in untreated *A. marina* membranes does not arise specifically from the fraction of centres lacking an intact Mn-cluster that are potentially responsible for the non-decaying fluorescence observed in *Figure 2*. (ii) The possibility that photosystem I (PSI) contributed to the light-induced $O_2$ consumption by reducing oxygen to $O_2^{•-}$ in membranes was tested (*Appendix 6—figure 3*). In the presence of DCMU, PSI-driven $O_2$ reduction mediated by methyl viologen only took place when exogenous electron donors to PSI were provided. This indicates that there is no contribution from PSI-induced $O_2$ reduction in *Figure 6A*, where exogenous PSI donors are absent. (iii) The higher $^1O_2$ production is also seen in *A. marina* cells compared to FR *C. thermalis* cells (*Appendix 6—figure 4A*), and thus is not an artefact associated with the isolation of membranes (e.g. damaged photosystems or free chlorophyll). WL *C. thermalis* cells also showed low levels of $^1O_2$ production, similar to those measured in membranes (*Appendix 6—figure 4B*). The reliability of the His-trapping method to monitor $^1O_2$ production in intact cyanobacterial cells has been previously demonstrated (*Rehman et al., 2013*).

*Figure 6B* shows the effect of 30 min of saturating illumination (red light) on the activity of the Chl-d-PSII, Chl-a-PSII and Chl-f-PSII. The results show that Chl-d-PSII is significantly more susceptible to light induced loss of activity compared to Chl-f-PSII, and to a lesser extent to Chl-a-PSII, and this can be correlated to the higher levels of $^1O_2$ production in Chl-d-PSII.

## Discussion

We investigated several functional properties of the two different types of far-red PSII, (i) the constitutive Chl-d-PSII of *A. marina*, and (ii) the facultative Chl-f-PSII of *C. thermalis*. We compared these properties with each other and with those of Chl-a-PSII, from either WL *C. thermalis*, *Synechocystis* or *T. elongatus*, looking for differences potentially related to the diminished energy available in the two long-wavelength PSII variants.

### Forward electron transfer and enzymatic activity

The turnover of the water oxidation cycle is comparably efficient in all three types of PSII, as shown by their near-identical flash patterns in thermoluminescence (*Appendix 2—figure 1*), $O_2$ release (*Figure 3*), and UV spectroscopy (*Figure 4*). In PSII, a photochemical 'miss factor' can be calculated from the damping of the flash patterns of $O_2$ evolution. These misses, which are typically ~10% in Chl-a-PSII, are mainly ascribed to the μs to ms recombination of $S_2TyrZ^•Q_A^-$ and $S_3TyrZ^•Q_A^-$ states (*Grabolle and Dau, 2007*). Despite the diminished energy available, the miss factors in both types of far-red PSII were virtually unchanged compared to Chl-a-PSII, which also suggests that the misses have the same origin. If so, the energy gaps between TyrZ and $P_{D1}$, and thus their redox potentials, would be essentially unchanged. These conclusions agree with those in earlier work on Chl-d-PSII (*Shevela et al., 2006*) and on Chl-f-PSII (*Nürnberg et al., 2018*).

The similar flash-patterns also indicate that, after the primary charge separation, the electron transfer steps leading to water oxidation must have very similar efficiencies in all three types of PSII, that is, close to 90%, and that there are no major changes affecting the kinetics of forward electron transfer. In the case of Chl-f-PSII, this confirms earlier suggestions based on flash-dependent thermoluminescence measurements (*Nürnberg et al., 2018*). Indeed, electron transfer from $Q_A^-$ to $Q_B$/$Q_B^-$, monitored by fluorescence, showed no significant differences in kinetics in the three types of PSII (*Figure 2A*).

### Back reactions and singlet oxygen production

The most striking difference between the three types of PSII is that the Chl-d-PSII of *A. marina* shows a decreased stability of $S_2Q_A^-$, indicated by the lower temperature of its TL peak and the correspondingly faster luminescent decay kinetics (*Figure 5*), and consequently a significant increase in $^1O_2$ generation under high light (*Figure 6A*). This likely corresponds to the decrease in the energy gap between Phe and $Q_A$ predicted to result from the ~100 meV lower energy available when using light

at ~720 nm to do photochemistry (**Nürnberg et al., 2018**; **Cotton et al., 2015**). This is also supported by the estimates in the literature of the redox potential ($E_m$) values of Phe/Phe$^-$ and $Q_A/Q_A^-$ in Mn-containing Chl-d-PSII: compared to Chl-a-PSII, the estimated increase of ~125 mV in the $E_m$ of Phe/Phe$^-$ is accompanied by an estimated increase of only ~60 mV in the $E_m$ of $Q_A/Q_A^-$, which implies that a normal energy gap between the excited state of the primary donor ($Chl_{D1}^*$) and the first and second radical pairs ($Chl_{D1}^+Phe^-$ and $P_{D1}^+Phe^-$) is maintained, but the energy gap between $P_{D1}^+Phe^-$ and $P_{D1}^+Q_A^-$ is significantly decreased (~325 meV vs ~385 meV) (**Allakhverdiev et al., 2011**). The changes in the D1 and D2 proteins of *A. marina* responsible for the changes in the $E_m$ of Phe/Phe$^-$ and $Q_A/Q_A^-$ are currently unknown.

Our results indicate that in Chl-d-PSII, the decrease in the energy gap between Phe and $Q_A$ favours charge recombination by the back-reaction route (via $P_{D1}^+Phe^-$), forming the reaction centre chlorophyll triplet state (**Rutherford et al., 1981**), which acts as an efficient sensitizer for $^1O_2$ formation (**Johnson et al., 1995**; **Vass and Cser, 2009**; **Keren et al., 1995**; **Keren et al., 2000**). Consequently, the Chl-d-PSII is more sensitive to high light (**Figure 6B**), reflecting the fact that this long-wavelength form of PSII has evolved in shaded epiphytic environments (**Nürnberg et al., 2018**; **Miyashita et al., 1996**; **Cotton et al., 2015**; **Davis et al., 2016**; **Murakami et al., 2004**; **Miller et al., 2005**; **Kühl et al., 2005**; **Mohr et al., 2010**). The increase in the proportion of recombination going via $P_{D1}^+Phe^-$ in Chl-d-PSII can also result in a higher repopulation of the excited state of the primary donor ($Chl_{D1}^*$), with a consequent increase in radiative decay (high luminescence).

In contrast to the Chl-d-PSII, the Chl-f-PSII shows no increased production of $^1O_2$ and no increased sensitivity to high light compared to Chl-a-PSII, in the conditions tested here (**Figure 6**). The back-reactions appear to be little different from the Chl-a-PSII except for the more stable (more slowly recombining) $S_2Q_A^-$, as seen by fluorescence (**Figure 2**) and luminescence (**Figure 5**) decay. These properties may seem unexpected because this type of PSII has the same energy available for photochemistry as the Chl-d-PSII. In the Chl-d-PSII the lower energy of $Chl_{D1}^*$ is matched by an increase in the $E_m$ of Phe/Phe$^-$. In the Chl-f-PSII of *C. thermalis* and of the other Chl-f containing species, the $E_m$ of Phe/Phe$^-$ is also expected to be increased by the presence, in the far-red D1 isoform, of the strong H-bond from Glu130 (**Figure 7**), which is characteristic of high-light D1 variants in cyanobacteria (**Sugiura et al., 2010**). In Chl-a-PSII, this change has been reported to induce an increase in the $E_m$ of Phe/Phe$^-$ between ~15 and~30 mV (**Sugiura et al., 2010**; **Merry et al., 1998**): an increase of this size would only partially compensate for the ~100 meV decrease in the energy of $Chl_{D1}^*$ in Chl-f-PSII, and this would result in a smaller energy gap between $Chl_{D1}^*$ and the first and second radical pairs $Chl_{D1}^+Phe^-$ and $P_{D1}^+Phe^-$. This would favour the repopulation of $Chl_{D1}^*$ by back-reaction from $P_{D1}^+Phe^-$ (even if the repopulation of $P_{D1}^+Phe^-$ from the $P_{D1}^+Q_A^-$ state did not increase), resulting in the higher luminescence of Chl-f-PSII, as proposed earlier (**Nürnberg et al., 2018**). Increased decay of the $P_{D1}^+Q_A^-$ radical pair via the radiative route could in principle decrease the decay via the triplet route, but the overall small yield of luminescence means that this could be a minor effect.

Additionally, the longer lifetime of $S_2Q_A^-$ recombination in Chl-f-PSII indicates that the $E_m$ of $Q_A/Q_A^-$ has increased to compensate the up-shift in the $E_m$ of Phe/Phe$^-$ and to maintain an energy gap between Phe and $Q_A$ large enough to prevent an increase in $P_{D1}^+Phe^-$ repopulation and thus in reaction centre chlorophyll triplet formation. This situation occurs in the PsbA3-D1 high light variant of *T. elongatus*, although the protein changes responsible for the increase in the $E_m$ of $Q_A/Q_A^-$ are not known (**Sugiura et al., 2010**). A slower $S_2Q_A^-$ recombination could also arise from an increase in the redox potential of $P_{D1}/P_{D1}^+$ (**Diner et al., 2001**), but this would likely compromise forward electron transfer in Chl-f-PSII by decreasing the driving force for stabilization of $Chl_{D1}^+Phe^-$ by the formation of $P_{D1}^+Phe^-$, if the redox potential of $Chl_{D1}/Chl_{D1}^+$ was not increased accordingly. If this problem were avoided by an equivalent increase in the redox potential of $Chl_{D1}/Chl_{D1}^+$, this would likely compromise charge separation, as a matching up-shift the $Chl_{D1}^+/Chl_{D1}^*$ couple would occur, thus decreasing the reductive power of $Chl_{D1}^*$. As a 720 nm pigment, the reducing power of $Chl_{D1}^*$ seems likely to be already compromised in Chl-f-PSII. The results here (**Figure 2A**) show forward electron transfer and charge separation appear to be comparable to Chl-a PSII and so it is unlikely that redox tuning of $P_{D1}$ and $Chl_{D1}$ is responsible for the stabilisation of the $S_2Q_A^-$ recombination .

```
A
  T. elong PsbA1      110 GPYQLIIFHFLLGASCYMGRQWELSYRLGMRPWI 143
  T. elong PsbA3      110 GPYQLIIFHFLIGVFCYMGREWELSYRLGMRPWI 143
  C. therm FR         111 GPYQMIGFHYIPALCCYAGREWELSYRLGMRPWI 144
  C. therm WL1        110 GPYQLVIFHFLIGCFCYMGRQWELSYRLGMRPWI 143
  C. therm WL2        110 GPYQLVIFHFLIGVFCYMGREWELSYRLGMRPWI 143
  C. therm WL3        110 GPYQLVIFHFLIGVFCYMGREWELSYRLGMRPWI 143
  A. marin 1          113 GPYQLIILHFLIAIWTYLGRQWELSYRLGMRPWI 146
  A. marin 2          110 GPYQLIIFHYMIGCICYLGRQWEYSYRLGMRPWI 143
  A. marin 3          110 GPYQLIIFHYMIGCICYLGRQWEYSYRLGMRPWI 143

B
  Leptol JSC-1        110 GPYQMIAAHYVPALCCYMGREWELSYRLGMRPWI 143
  Oscill JSC-12       111 GPYQMIGAHYIPALACYMGRQWELSYRLGMRPWI 144
  Caloth NIES-267     110 GPYQMIAFHYIPALSCYMGREWELSYRLGMRPWI 143
  Mastigo BC008       111 GPYQMIAFHYIPALACYMGREWELSYRLGMRPWI 144
  C. therm FR         111 GPYQMIGFHYIPALCCYAGREWELSYRLGMRPWI 144
  Caloth PCC7507      111 GPYQMIAFHYIPALSCYMGREWELSYRLGMRPWI 144
  Caloth NIES-3974    111 GPYQMIAFHYIPALACYMGREWELSYRLGMRPWI 144
  Fische NIES-592     111 GPYQMIGFHYIPALACYMGREWELSYRLGMRPWI 144
  Fische NIES-3754    111 GPYQMIGFHYIPALACYMGREWELSYRLGMRPWI 144
  Mastigo SAG4.84     111 GPYQMIGFHYIPALACYMGREWELSYRLGMRPWI 144
  Chlorog PCC6912     111 GPYQMIGFHYIPALACYMGREWELSYRLGMRPWI 144
  Fische PCC9605      111 GPYQMIGFHYIPALACYMGREWELSYRLGMRPWI 144
  Halomicr. Hongd.    110 GPYQMIAFHYIPALLCYMGREWELSYRLGMRPWI 143
  Synechoco PCC7335   109 GPYQMIAFHYIPALLCYLGREWELSYRLGMRPWI 142
  Pleuroc CCALA161    110 GPYQMIAFHYIPALCCYLGREWELSYRLGMRPWI 143
  Hydroco NIES-593    110 GPYQMIALHYVPALCCYLGREWELSYRLGMRPWI 143
  Pleuroc PCC7327     110 GPYQMIALHYVPALCCYLGREWELSYRLGMRPWI 143
```

**Figure 7.** Occurrence of the high light-associated D1-Q130E substitution in the different types of PSII. (**A**) Multi-alignment of the D1 proteins of *T. elongatus*, *C. thermalis* and *A. marina*. The Q130E substitution is present also in the far-red light-induced D1 isoform of *C. thermalis* (*C. therm* FR) and in two out of three of its non-far-red induced D1 isoforms (*C. therm* WL2 and 3) but is not present in any of the three D1 isoforms of *A. marina*. (**B**) Multi-alignment of the far-red light induced D1 isoforms of *C. thermalis* and other Chl-f species. The presence of E130 is conserved in the far-red light induced D1 isoforms of most of the cyanobacteria species capable of far-red light photo-acclimation. Both alignments were done using Clustal Omega (*Sievers et al., 2011*), the sequences were retrieved from the KEGG (https://www.kegg.jp/) and NCBI (https://www.ncbi.nlm.nih.gov/) databases. For each alignment only a 33 amino acid region is shown, the start and end positions with respect to each full sequence are indicated with numbers. The Q130E substitution is highlighted as white font on black background. The far-red D1 sequence from *C. thermalis* is framed in red. All sequences used for the multi-alignments are provided in *Figure 7—source data 1*.

The online version of this article includes the following source data for figure 7:

**Source data 1.** D1 sequences used for multi-alignments.

## Effects of the pigment composition on the energetics of the far-red PSII

In addition to changes in the redox potentials of Phe and $Q_A$, the size and pigment composition of the antennas of Chl-d-PSII and Chl-f-PSII could also contribute to the functional differences reported in the present work. These differences are summarized in *Figure 8*.

In PSII, two factors will determine the yield of charge separation: (i) the relative population of the excited state of the primary donor, $Chl_{D1}^*$, which depends on the dynamics of excitation energy transfer between pigments, and (ii) the rate of population of the second radical pair, $P_{D1}^+Phe^-$, that is more stable (less reversible) than the first radical pair, $Chl_{D1}^+Phe^-$. This rate is determined by the rates

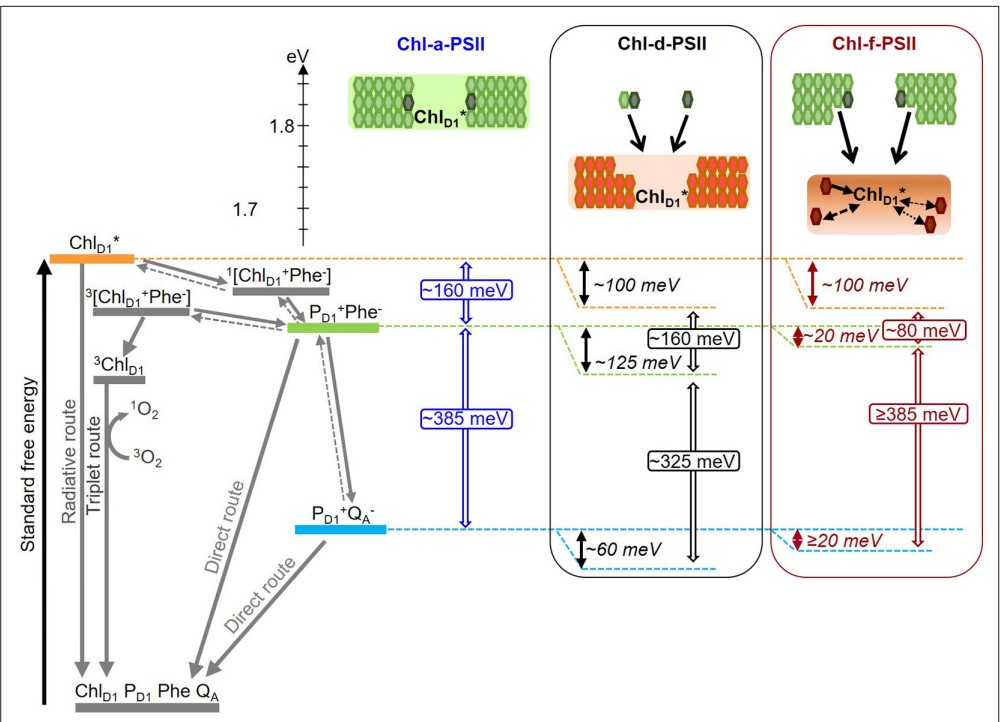

**Figure 8.** Model of the energy differences in Chl-a-PSII, Chl-d-PSII and Chl-f-PSII. The top part of the figure represents the localization of the excitation energy over the antenna pigments and Chl$_{D1}$* (energies in eV, scale on the left side). The localization of the excitation energy is indicated by the coloured boxes (green for Chl-a, orange for Chl-d and brown for Chl-f), without necessarily assuming a full equilibration (see main text). In Chl-a-PSII, the excitation is distributed over Chl$_{D1}$, 34 antenna Chl-a (light green) and 2 pheophytin a (Phe-a, dark grey); in Chl-d-PSII, the excitation is distributed over Chl$_{D1}$ and 31 antenna Chl-d (orange) but not over the 1 Chl-a and 2 Phe-a, that transfer excitation downhill to the Chl-d pigments (black arrows); in Chl-f-PSII, the excitation is distributed only over Chl$_{D1}$ and 4 antenna Chl-f (brown), while the remaining 29 Chl-a and 2 Phe-a transfer excitation energy downhill to the far-red pigments. In Chl-f-PSII, 3 of the far-red antenna pigments are at longer wavelength than Chl$_{D1}$, so transfer of excitation energy from them to Chl$_{D1}$ is less efficient (dashed and dotted black arrows, representing the possible heterogeneity in excitation energy transfer kinetics). The grading of the coloured box for Chl-f represents uncertainty in the degree of excited state sharing between the longest wavelength chlorophylls and Chl$_{D1}$. The bottom part of the figure represents, on the left, the energetics of the radical pairs and the recombination routes in PSII (direct route: via electron tunnelling; triplet route: via the formation of the triplet state of the primary electron donor; radiative route: via luminescence emission), with the electron transfer steps between P$_{D1}$ and the Mn-cluster omitted for clarity. The solid and dashed grey arrows represent exergonic and endergonic electron transfer, respectively. The horizontal dashed lines represent the standard free energies of Chl$_{D1}$* (orange), P$_{D1}$+Phe- (light green) and P$_{D1}$+Q$_A$- (light blue). The P$_{D1}$+Phe- radical pair, when formed from the backreaction of P$_{D1}$+Q$_A$-, can be either in the singlet state or in the triplet state. $^1$[P$_{D1}$+Phe-] recombines via $^1$[Chl$_{D1}$+Phe-] and Chl$_{D1}$*, while $^3$[P$_{D1}$+Phe-] recombines via $^3$[Chl$_{D1}$+Phe-] and $^3$Chl$_{D1}$. The free energy gaps between Chl$_{D1}$* and P$_{D1}$+Phe- and between P$_{D1}$+Phe- and P$_{D1}$+Q$_A$- in Chl-a-PSII (blue) and our current estimates for Chl-d-PSII (black) and Chl-f-PSII (dark red) are represented on the right.

of the primary charge separation (forming Chl$_{D1}$+Phe-) and of its stabilization by secondary electron transfer (forming P$_{D1}$+Phe-), and hence by the energetics of these electron transfer steps.

In the Chl-a-PSII core, the 37 chlorins absorb light between ~660 and ~690 nm and are therefore almost isoenergetic to the Chl$_{D1}$ primary donor absorbing at 680 nm. Given the small energy differences, there is little driving force for downhill 'funnelling' of excitation energy to Chl$_{D1}$, making it a 'shallow trap'. Different models have been proposed to explain the shallowness of the photochemical trap in Chl-a-PSII.

In the trap-limited model, the transfer of excitation between pigments is significantly faster than the electron transfer reactions leading to P$_{D1}$+Phe- formation, and a near-complete equilibration of the excitation energy is established over all pigments, including Chl$_{D1}$, with a distribution that is

**Table 4.** Excitation energy partitions calculated for the three types of PSII assuming excitation equilibration between the pigments.
E$^*$ denotes the energy of the excited state, obtained by applying a+5 nm Stoke's shift to the absorption of the pigments, n is the number of pigments belonging to each state and Pi is the normalized partition of the excited states, calculated following Boltzmann distribution (**Laible et al., 1994**).

Chl-a-PSII

| State | E$^*$ | n | Pi |
|---|---|---|---|
| Bulk Chl-a/Phe-a | 685 | 34 | 0.878 |
| Chl$_{D1680}$ | 685 | 1 | **0.026** |
| F$_{685}$ | 685 | 1 | 0.026 |
| F$_{695}$ | 695 | 1 | 0.071 |

Chl-d-PSII

| State | E$^*$ | n | Pi |
|---|---|---|---|
| Chl-a/Phe-a | 685 | 3 | 0.002 |
| Chl$_{D1720}$ | 725 | 1 | **0.029** |
| Bulk Chl-d | 725 | 33 | 0.969 |

Chl-f-PSII

| State | E$^*$ | n | Pi |
|---|---|---|---|
| Bulk Chl-a/Phe-a | 685 | 32 | 0.046 |
| Chl$_{D1721}$ | 726 | 1 | **0.075** |
| F$_{720}$/A$_{715}$ | 720 | 1 | 0.043 |
| F$_{731}$/A$_{726}$ | 731 | 1 | 0.117 |
| F$_{737}$/A$_{732}$ | 737 | 1 | 0.2 |
| F$_{748}$/A$_{743}$ | 748 | 1 | 0.52 |

The states are denoted as follows: Chl$_{D1}$ is the primary donor (Pi highlighted in bold), Bulk indicates the antenna pigments considered as isoenergetic, and F indicates the antenna pigments considered separately from the bulk, with the fluorescence emission wavelength indicated. In the case of the far-red pigments in Chl-f-PSII the peak absorptions (A) are also indicated, as taken from **Judd et al., 2020**.

determined by their individual site energies (**van Mieghem et al., 1995**; **Schatz et al., 1987**; **Dau and Sauer, 1996**). This leads to a low population of Chl$_{D1}$$^*$ (**Table 4**), that is diminished as a function of the number of quasi-isoenergetic pigments with which it shares the excitation energy.

In the transfer-to-trap limited model, the small driving force for downhill 'funnelling' of excitation energy to Chl$_{D1}$ causes kinetic bottlenecks for excitation energy equilibration between the core antenna complexes CP43 and CP47 and for excitation energy transfer from these antennas to the reaction centre (**Jennings et al., 2000**; **Pawlowicz et al., 2007**; **Raszewski and Renger, 2008**; **Kaucikas et al., 2016**). In this scenario, there is not a full equilibration of the excitation energy over all pigments, but the relatively slow and reversible energy transfer from the core antennas to the reaction centre still leads to a relatively low population of Chl$_{D1}$$^*$.

Irrespectively of the differences in the details of the kinetic limitation to photochemical trapping between the two models, the common requirement for establishing a high quantum yield of charge separation is a sufficiently large overall energy gap (~160 meV, ) between Chl$_{D1}$$^*$ and P$_{D1}$$^+$Phe$^-$, that comprises the primary charge separation (Chl$_{D1}$$^*$ ↔ Chl$_{D1}$$^+$Phe$^-$) and secondary electron transfer (Chl$_{D1}$$^+$Phe$^-$ ↔ P$_{D1}$$^+$Phe$^-$), as shown in **Figure 8**. An energy gap of this magnitude is required to avoid rapid recombination to the excited state Chl$_{D1}$$^*$, thereby limiting the probability of its dissipation via non-photochemical relaxation to the ground state in the antenna (**Raszewski and Renger, 2008**; **Schatz et al., 1988**).

For Chl-d-PSII the antenna system is comparable to that in Chl-a-PSII: all 34 Chl-d molecules, including the primary donor Chl$_{D1}$ at ~720 nm, are close in wavelength and thus both systems are expected to have comparable Chl$_{D1}$$^*$ population (**Table 4**), irrespective of the rate-limitation model assumed. Chl-a-PSII and Chl-d-PSII should therefore have the same energetic requirements to ensure a sufficiently high yield of charge separation. Given that the energy of Chl$_{D1}$$^*$ is ~100 meV lower in Chl-d-PSII than in Chl-a-PSII, the energy level of the second and more stable radical pair, P$_{D1}$$^+$Phe$^-$, needs to be decreased by ~100 meV in Chl-d-PSII relative to Chl-a-PSII. This corresponds to the published E$_m$ of Phe/Phe$^-$ (**Allakhverdiev et al., 2011**) and to the kinetic data (**Figures 5 and 6**), as detailed in the previous section.

In *A. marina* membranes, additional Chl-d containing antenna proteins, which form supercomplexes with PSII cores, have been reported to increase the Chl-d-PSII antenna size by almost 200% (**Chen et al., 2005**). This will likely result in an increased sharing of the excited state, leading to a diminished population of Chl$_{D1}$$^*$, and thus a bigger requirement for an energy drop between Chl$_{D1}$$^*$ and P$_{D1}$$^+$Phe$^-$ to ensure efficient charge separation. At the same time, the larger near-isoenergetic antenna

could also contribute to its higher luminescence, by increasing the probability of $Chl_{D1}^{*}$ decay via radiative emission with respect to photochemical re-trapping (*Rappaport and Lavergne, 2009*). This is similar to what happens in plant PSII, where the yield of photochemical trapping of excitation energy is decreased by 10–15% by the association of the Light Harvesting Complex antennas (*Engelmann et al., 2005*).

The pigment layout of Chl-f-PSII is very different from that of Chl-a-PSII and Chl-d-PSII. The 30 Chl-a molecules are energetically separated from $Chl_{D1}$, absorbing at 720 nm, by >30 nm (>3 $k_BT$). This means the excitation energy resides predominantly on $Chl_{D1}^{*}$ and on the other 4 far-red pigments. If the equilibration of the excitation energy between the 5 far-red pigments were significantly faster than charge separation, this pigment arrangement would result in a higher probability of populating $Chl_{D1}^{*}$ in Chl-f-PSII than in Chl-a-PSII and Chl-d-PSII (*Table 4*). The higher $Chl_{D1}^{*}$ population in Chl-f-PSII could ensure that sufficient yield of charge separation is achieved even when the $E_m$ of Phe/Phe$^-$ is increased by much less that the 100 meV needed to compensate for the nominally lower energy in $Chl_{D1}^{*}$.

However, thermal equilibration of the excitation energy over the entire antenna in Chl-f-PSII might not occur due to 3 of the 4 long-wavelength antenna chlorophylls absorbing at longer wavelength than $Chl_{D1}$. This type of antenna energetics could result in rapid excited state equilibration in each of the three main pigment-protein complexes (CP43, CP47 and reaction centre), due to rapid energy transfer from Chl-a to Chl-f/d (with visible light excitation) followed by slower transfer from the two postulated far-red antenna pools to $Chl_{D1}$, leading to a transfer-to-trap limited bottleneck. As a result, the kinetics of excitation energy transfer from the red and far-red antenna to the reaction centre could be more complex than in Chl-a-PSII and Chl-d-PSII, explaining the spread in charge separation kinetics that has been suggested based on ultrafast absorption data (*Zamzam et al., 2020*) and the slower excitation energy trapping kinetics measured by time-resolved fluorescence (*Mascoli et al., 2020*).

The driving force for charge separation is decreased in Chl-f-PSII also by the smaller energy gap between $Chl_{D1}^{*}$ and $P_{D1}^{+}Phe^-$, estimated to be ~80 meV in Chl-f-PSII compared to ~160 meV in Chl-a-PSII and Chl-d-PSII. This decrease in the energy gap between $Chl_{D1}^{*}$ and $P_{D1}^{+}Phe^-$ is necessary in Chl-f-PSII to avoid the increased photosensitivity seen in Chl-d-PSII by maintaining a large energy gap between $P_{D1}^{+}Phe^-$ and $P_{D1}^{+}Q_A^-$ (~385 meV) (*Figure 8*). Nonetheless, the slower excitation energy transfer and the smaller energy gap between $Chl_{D1}^{*}$ and $P_{D1}^{+}Phe^-$ could be partially compensated by the decreased dilution of the excitation energy on $Chl_{D1}^{*}$ arising from the small number of long-wavelength antenna pigments, resulting in only a small loss of trapping efficiency (*Mascoli et al., 2020*) and a near-negligible effect on enzyme turnover efficiency (*Figures 2–5*).

This energetic balancing trick in Chl-f-PSII, which allows both reasonably high enzyme efficiency and high resilience to photodamage (by limiting recombination via the repopulation of $P_{D1}^{+}Phe^-$) despite working with 100 meV less energy, comes with a very significant disadvantage: its absorption cross-section at long wavelength is ~7 times smaller than that of the Chl-a-PSII core antenna in visible light. In the case of Chl-f-PSII, evolution therefore seems to have prioritized the minimization of harmful charge recombination, by maintaining a big energy gap between Phe and $Q_A$, over light collection and photochemical quantum efficiency. This makes sense as this system has evolved as a facultative survival mechanism, that is not advantageous when visible light is available. It must be noted that *in vivo* the far-red antenna cross-section of Chl-f-PSII is increased by the presence of red-shifted phycobilisomes, that replace the visible light-absorbing phycobilisomes when the cells are adapted to far-red light (*Gan et al., 2014*).

In contrast, Chl-d-PSII seems to have maximized light collection at long wavelengths (with its full-size far-red antenna) and maximized the yield of charge separation (by maintaining the full $Chl_{D1}^{*}$ to $P_{D1}^{+}Phe^-$ driving force). However, the energy shortfall at long wavelength is lost from the 'energy headroom' (mainly from the transmembrane energy gap between Phe and $Q_A$) that is proposed to minimize harmful charge recombination by buffering the effects of pulses of the trans-membrane electric field associated with fluctuations in light intensity (*Davis et al., 2016*; *Davis et al., 2017*). This seems to correspond well to the shaded and stable epiphytic niche that *A. marina* occupies (*Nürnberg et al., 2018*; *Miyashita et al., 1996*; *Cotton et al., 2015*; *Davis et al., 2016*; *Murakami et al., 2004*; *Miller et al., 2005*; *Kühl et al., 2005*; *Mohr et al., 2010*).

Chl-d-PSII and Chl-f-PSII have evolved different strategies to do oxygenic photosynthesis in far-red light and have been impacted differently by the decrease in energy available. Understanding how the redox tuning of the electron transfer cofactors and the layout of the far-red pigments determine the

trade-off between efficiency and resilience in PSII is a necessary step to inform strategies aimed at using far-red photosynthesis for agricultural and biotechnological applications.

The present findings suggest the exchange of the full Chl-a manifold to long-wavelength chlorophylls, as seen in Chl-d-PSII (*A. marina*), should allow efficient oxygenic photosynthesis, but only under constant shading and stable light conditions: for example, for cultivation under LED light (vertical farming, etc). The more robust, facultative Chl-f PSII has an intrinsically low absorption cross-section in the far red; however, this could be beneficial in a shaded canopy, especially in combination with a suitably designed far-red external antenna.

# Materials and methods

## Key resources table

| Reagent type (species) or resource | Designation | Source or reference | Identifiers | Additional information |
|---|---|---|---|---|
| Strain, strain background (*Chroococcidiopsis*) | *Chroococcidiopsis thermalis* sp. PCC7203 | Pasteur Culture Collection of Cyanobacteria | NCBI:txid251229 | |
| Strain, strain background (*Acaryochloris*) | *Acaryochloris marina MBIC 11017* | Marine Biotechnology Institute Culture Collection | NCBI:txid155978 | |
| Strain, strain background (*Synechocystis*) | *Synechocystis* sp. PCC6803 | Pasteur Culture Collection of Cyanobacteria | NCBI:txid1148 | Glucose tolerant |
| Genetic reagent (*Thermosynechococcus elongatus* BP-1) | *ΔpsbA1, ΔpsbA2* | DOI:10.1016 /j. bbabio.2008.01.007 | WT*3 | |
| Chemical compound, drug | MES (2-(N-morpholino)ethanesulfonic acid) | Thermo Scientific | J18886.A1 | |
| Chemical compound, drug | β-DM (n-Dodecyl-β-D-maltoside) | Thermo Scientific | 89,903 | |
| Chemical compound, drug | DCMU (3-(3,4-dichlorophenyl)–1,1-dimethylurea) | Sigma-Aldrich | D2425 | |
| Chemical compound, drug | DCBQ (2,5-Dichloro-1,4-benzoquinone) | Sigma-Aldrich | 431,982 | |
| Chemical compound, drug | potassium ferricyanide | Sigma-Aldrich | 244,023 | |
| Chemical compound, drug | L-Histidine | BioChemica | A3738 | |
| Chemical compound, drug | sodium azide ($NaN_3$) | Sigma-Aldrich | S2002 | |
| Chemical compound, drug | Methyl viologen dichloride hydrate | Sigma-Aldrich | 856,177 | |
| Chemical compound, drug | TMPD (N,N,N',N'-tetramethyl-p-phenylenediamine) | Sigma-Aldrich | T3134 | |
| Chemical compound, drug | Rose Bengal | Sigma-Aldrich | 330,000 | |
| Chemical compound, drug | PPBQ (Phenyl-p-benzoquinone) | Sigma-Aldrich | PH005156 | |
| Chemical compound, drug | 6-Aminocaproic acid | Sigma-Aldrich | A7824 | |
| Chemical compound, drug | Benzamidine hydrochloride hydrate | Alfa Aesar | J62823 | |

## Cyanobacterial growth

*Acaryochloris marina* was grown in a modified liquid K-ESM medium containing 14 µM iron (**Bailleul et al., 2008**), at 30 °C under constant illumination with far-red light (750 nm, Epitex; L750-01AU) at ~30 µmol photons m$^{-2}$ s$^{-1}$. *Chroococcidiopsis thermalis* PCC7203 was grown in liquid BG11 medium (**Stanier et al., 1979**) at 30 °C, under two illumination conditions: white light at ~30 µmol photons m$^{-2}$ s$^{-1}$ (for WL *C. thermalis* samples) and far-red light (750 nm, Epitex; L750-01AU) at ~30 µmol photons m$^{-2}$ s$^{-1}$ (for FR *C. thermalis* samples). *Synechocystis sp.* PCC6803 was grown in liquid BG11 medium at 30 °C under constant illumination with white light at ~30 µmol photons m$^{-2}$ s$^{-1}$. The *Thermosynechococcus elongatus ΔpsbA1, ΔpsbA2* deletion mutant (WT*3, **Sugiura et al., 2008**) was grown in liquid DNT medium at 45 °C.

## Isolation of membranes and PSII cores

Membranes were prepared as described in Appendix 7, frozen in liquid nitrogen and stored at –80 °C until use. Partially purified *C. thermalis* PSII cores retaining oxygen evolution activity were isolated by anion exchange chromatography using a 40 ml DEAE column. The column was equilibrated with 20 mM MES (2-(*N*-morpholino)ethanesulfonic acid)-NaOH pH 6.5, 5 mM $CaCl_2$, 5 mM $MgCl_2$ and 0.03% (w/v) β-DM (n-Dodecyl-β-D-maltoside) and elution was done using a linear gradient of $MgSO_4$ from 0 to 200 mM in 100 min (in the same buffer conditions as those used to equilibrate the column), with a flow rate of 4 ml $min^{-1}$. Fractions enriched in PSII were pooled, frozen in liquid nitrogen and stored at –80 °C. PSII-PsbA3 cores from *T. elongatus* WT*3 were purified as previously described (*Sugiura et al., 2014*).

## Fluorescence

Flash-induced chlorophyll fluorescence and its subsequent decay were measured with a fast double modulation fluorimeter (FL 3000, PSI, Czech Republic). Excitation was provided by a saturating 70 µs flash at 630 nm and the decay in $Q_A^-$ concentration was probed in the 100 µs to 100 s time region using non-actinic measuring pulses following a logarithmic profile as described in *Vass et al., 1999*. The first measuring point was discarded during the data analysis because it contains a light artefact arising from the tail of the saturating flash used for excitation. Details on the analysis of the fluorescence curves (based on *Crofts and Wraight, 1983*; *Vass et al., 1999*; *Cser et al., 2008*) are provided in Appendix 7. Membrane samples were adjusted to a total chlorophyll concentration of 5 µg Chl $ml^{-1}$ in resuspension buffer, pre-illuminated with room light (provided by a white fluorescent lamp, ~80 µmol photons $m^{-2}$ $s^{-1}$) for 10 s and then kept in the dark on ice until used for measurements. Measurements were performed at 20 °C. Where indicated, 20 µM DCMU (3-(3,4-dichlorophenyl)–1,1-dimethylurea) was used.

## Thermoluminescence and luminescence

Thermoluminescence curves and luminescence decay kinetics were measured with a laboratory-built apparatus, described in *De Causmaecker, 2018*. Membrane samples were diluted in resuspension buffer to a final concentration of 5 µg Chl $ml^{-1}$ in the case of *A. marina* and FR *C. thermalis* and of 10 µg $ml^{-1}$ in the case of WL *C. thermalis* and *Synechocystis*. The samples were pre-illuminated with room light (provided by a white fluorescent lamp,~80 µmol photons $m^{-2}$ $s^{-1}$) for 10 s and then kept in the dark on ice for at least one hour before the measurements. When used, 20 µM DCMU was added to the samples before the pre-illumination step. Excitation was provided by single turnover saturating laser flashes (Continuum Minilite II, frequency doubled to 532 nm, 5 ns FWHM). Details on the measurement conditions and on the analysis of the luminescence decay kinetics are provided in Appendix 7.

## Oxygen evolution and consumption rates

Oxygen evolution and consumption rates were measured with a Clark-type electrode (Oxygraph, Hansatech) at 25 °C. Membrane samples were adjusted to a total chlorophyll concentration of 5 µg Chl $ml^{-1}$. Illumination was provided by a white xenon lamp filtered with a heat filter plus red filter, emitting 600–700 nm light at 7100 µmol photons $m^{-2}$ $s^{-1}$ (Quantitherm light meter, Hansatech). When required, the light intensity was reduced by using neutral density filters (Thorlabs). For PSII maximal oxygen evolution rates, 1 mM DCBQ (2,5-Dichloro-1,4-benzoquinone) and 2 mM potassium ferri-cyanide were used as an electron acceptor system. Photoinhibitory illumination was performed at room temperature for 30 min with a laboratory-built red LED (660 nm, 2600 µmol photons $m^{-2}$ $s^{-1}$). For histidine-mediated chemical trapping of singlet oxygen, 20 µM DCMU, 5 mM L-Histidine and, where specified, 10 mM sodium azide ($NaN_3$) were used. PSI activity was measured as the rate of oxygen consumption in presence of 20 µM DCMU and 100 µM methyl viologen using 5 mM sodium ascorbate and 50 µM TMPD (N,N,N′,N′-tetramethyl-p-phenylenediamine) as electron donors. The dye Rose Bengal was used at a final concentration of 0.1 µM. After all necessary additions, samples were left to equilibrate with air in the measuring chamber for 1 minute, under stirring, to start with similar dissolved $O_2$ concentrations in all measurements. For each measurement, a dark baseline was recorded for 1 min before starting the illumination.

## Flash-dependent oxygen evolution with Joliot electrode

Time-resolved oxygen polarography was performed using a custom-made centrifugable static ring-disk electrode assembly of a bare platinum working electrode and silver-ring counter electrode, as previously described (*Dilbeck et al., 2012*). For each measurement, membranes equivalent to 10 µg of total chlorophyll were used. Three different light sources were used to induce the S-state transitions: a red LED (613 nm), a far-red LED (730 nm) and a Xenon flashlamp. Details on the experimental setup and on the lights used are provided in Appendix 7. Measurements were performed at 20 °C. For each measurement, a train of 40 flashes fired at 900ms time interval was given and the flash-induced oxygen-evolution patterns were taken from the maximal $O_2$ signal of each flash and fitted with an extended Kok model with flash-independent miss factor, as described in *Nöring et al., 2008*.

## UV transient absorption

UV pump-probe absorption measurements were performed using a lab-built Optical Parametric Oscillator (OPO)-based spectrophotometer (*Joliot et al., 1998*) with a time resolution of 10 ns and a spectral resolution of 2 nm (see Appendix 7 for details on the setup). $\Delta I/I$ stands for differential absorption, a method that measures the changes in absorption depending on whether or not a sample is subjected to actinic light. Samples were diluted in resuspension buffer to a final concentration of 25 µg Chl ml$^{-1}$ for isolated *C. thermalis* and *T. elongatus* PSII cores and 40 µg Chl ml$^{-1}$ for *A. marina* membranes. Samples were pre-illuminated with room light (provided by a white fluorescent lamp,~80 µmol photons m$^{-2}$ s$^{-1}$) for 10 s and then kept in the dark on ice for at least one hour before the measurements. 100 µM PPBQ (Phenyl-p-benzoquinone) was added just before the measurements. The sample was refreshed between each train of flashes. For each measurement, a train of 20 flashes (6 ns FWHM) fired at 300ms time interval was given, and absorption changes measured at 100ms after each flash. The data were fitted according to *Lavergne, 1991*; *Lavorel, 1976*.

## Acknowledgements

This work was supported by BBSRC grants BB/R001383/1, BB/V002015/1 and BB/R00921X. JS acknowledges funding from the Labex Dynamo (ANR-11-LABX-0011–01). AB was in part supported by the French Infrastructure for Integrated Structural Biology (FRISBI) ANR-10-INBS-05. SS acknowledges support from Fondazione Cariplo, project "Cyanobacterial Platform Optimised for Bioproductions" (ref. 2016–0667).

## Additional information

### Funding

| Funder | Grant reference number | Author |
| --- | --- | --- |
| Biotechnology and Biological Sciences Research Council | BB/R001383/1 | Stefania Viola |
| Biotechnology and Biological Sciences Research Council | BB/V002015/1 | Stefania Viola |
| Biotechnology and Biological Sciences Research Council | BB/R00921X | Stefania Viola |
| Labex | ANR-11-LABX-0011-01 | Julien Sellés |
| French Infrastructure for Integrated Structural Biology | ANR-10-INBS-05 | Alain Boussac |
| Fondazione Cariplo | 2016-0667 | Stefano Santabarbara |
| Deutsche Forschungsgemeinschaft | NU 421/1 | Dennis Nürnberg |

| Funder | Grant reference number | Author |
|---|---|---|
| Deutsche Forschungsgemeinschaft | CRC1078- project A4 | Holger Dau |
| Deutsche Forschungsgemeinschaft | EXC2008/1 – 390540038 | Holger Dau |

The funders had no role in study design, data collection and interpretation, or the decision to submit the work for publication.

### Author contributions

Stefania Viola, Conceptualization, Data curation, Formal analysis, Investigation, Writing – original draft, Writing – review and editing; William Roseby, Investigation; Stefano Santabarbara, Data curation, Formal analysis, Writing – review and editing; Dennis Nürnberg, Ricardo Assunção, Data curation, Formal analysis, Investigation; Holger Dau, Resources, Formal analysis; Julien Sellés, Formal analysis, Investigation; Alain Boussac, Formal analysis, Investigation, Writing – review and editing; Andrea Fantuzzi, Conceptualization, Formal analysis, Funding acquisition, Project administration, Writing – review and editing; A William Rutherford, Conceptualization, Formal analysis, Funding acquisition, Writing – original draft, Project administration, Writing – review and editing

### Author ORCIDs

Stefania Viola ⬡ http://orcid.org/0000-0003-0773-8158
A William Rutherford ⬡ http://orcid.org/0000-0002-3124-154X

### Decision letter and Author response

Decision letter https://doi.org/10.7554/eLife.79890.sa1
Author response https://doi.org/10.7554/eLife.79890.sa2

## Additional files

### Supplementary files
• MDAR checklist

### Data availability
All data generated and analysed during this study have been included in the manuscript and supporting file and provided as source data files.

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

## Appendix 1

### Fluorescence decay kinetics in *Synechocystis* and WL *C. thermalis*

The fluorescence decay kinetics measured here in *Synechocystis* membranes (***Appendix 1—figure 1***), as well as those measured in *A. marina* and *C. thermalis* membranes (***Figure 2***), are slower than those measured in *Synechocystis* intact cells in a previous work (***Cser and Vass, 2007***). Additionally, a study of fluorescence decay times was previously reported comparing $Q_A^-$ lifetimes in *A. marina* and *Synechocystis* but in cells rather than membranes. In *A. marina* cells, the forward ($Q_A^-$ to $Q_B$) electron transfer rate was slower than in *Synechocystis* cells, while the $S_2Q_A^-$ recombination rate *A. marina* cells was faster than in *Synechocystis* cells (***Cser et al., 2008***). In both organisms, the fluorescence decay kinetics were faster than the values measured here in membranes. The faster rates in cells compared to isolated membranes are intrinsic to the type of sample used. The transmembrane electric field, which is present in cells but not in isolated membranes, is known to accelerate $Q_A^-$ decay in presence of DCMU (***Joliot and Joliot, 1980***). Additionally, the faster rates for $Q_A^-$ to $Q_B$ electron transfer in cells may be attributed to the $Q_B$ site in living cells functioning optimally at higher pH rather than at the pH 6.5 used here to maintain PSII donor-side function.

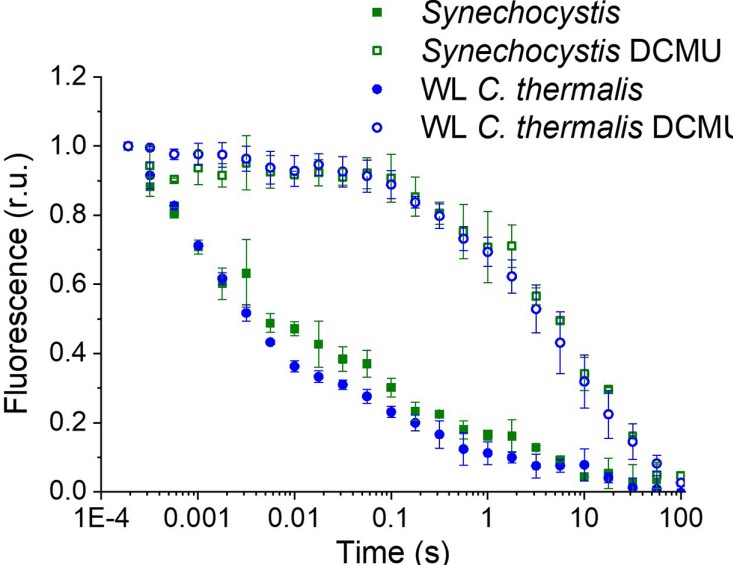

**Appendix 1—figure 1.** Fluorescence decay kinetics after a short saturating light pulse in isolated membranes of *Synechocystis* and WL *C. thermalis* in absence and presence of DCMU. The WL *C. thermalis* data are the same as those in ***Figure 2*** and ***Figure 2—figure supplement 1***. The *Synechocystis* datapoints represent the averages of two biological replicates,±s.d. (provided in ***Appendix 1—figure 1—source data 1***). All traces are normalized on the initial variable fluorescence ($F_m$-$F_0$, with $F_m$ measured 190 µs after the saturating flash).

The online version of this article includes the following source data for appendix 1—figure 1:

**Appendix 1—figure 1—source data 1.** Fluorescence decay kinetics Synechocystis.

## Appendix 2

### Flash dependence of thermoluminescence

*Appendix 2—figure 1* shows the TL emission after a series of saturating flashes in *A. marina* (panels **A**, **D** and **G**), WL *C. thermalis* (panels **B**, **E** and **H**) and FR *C. thermalis* (panels **C**, **F** and **I**) membranes. Although no major differences in the flash patterns could be observed between the three samples, the flash dependence of the TL peak intensities (panels **A**, **B** and **C**) and their peak temperatures (panels **D**, **E** and **F**) showed variability between biological replicates. Representative TL glow curves obtained in one biological replicate for each sample after 1–6 flashes are shown in panels **G**, **H** and **I**. The differences in the flash patterns between replicates are easily explained by some variability in both the $S_0/S_1$ and $Q_B/Q_B^-$ ratios present in the dark before the first flash, although the flash patterns suggest the presence of ~75% $S_1$/~25% $S_0$ and ~50% $Q_B$/~50% $Q_B^-$, based on *Rutherford et al., 1981*.

For WL *C. thermalis*, the smaller TL amplitude makes the peak temperature more difficult to estimate very precisely. For FR *C. thermalis*, a progressive broadening of the TL peak with increasing flash number made quantification less reliable, and for *A. marina* an increase in the baseline at high temperatures (also occurring to a smaller extent but still visible in WL *C. thermalis*) added to the difficulties in estimating the area of the TL peaks. For these reasons, the TL data are not precise enough to quantify potential differences in the S-state turnover efficiency in the different types of PSII, although they show that any such differences, if present, must be small (from the data in *Appendix 2—figure 1A, B and C*).

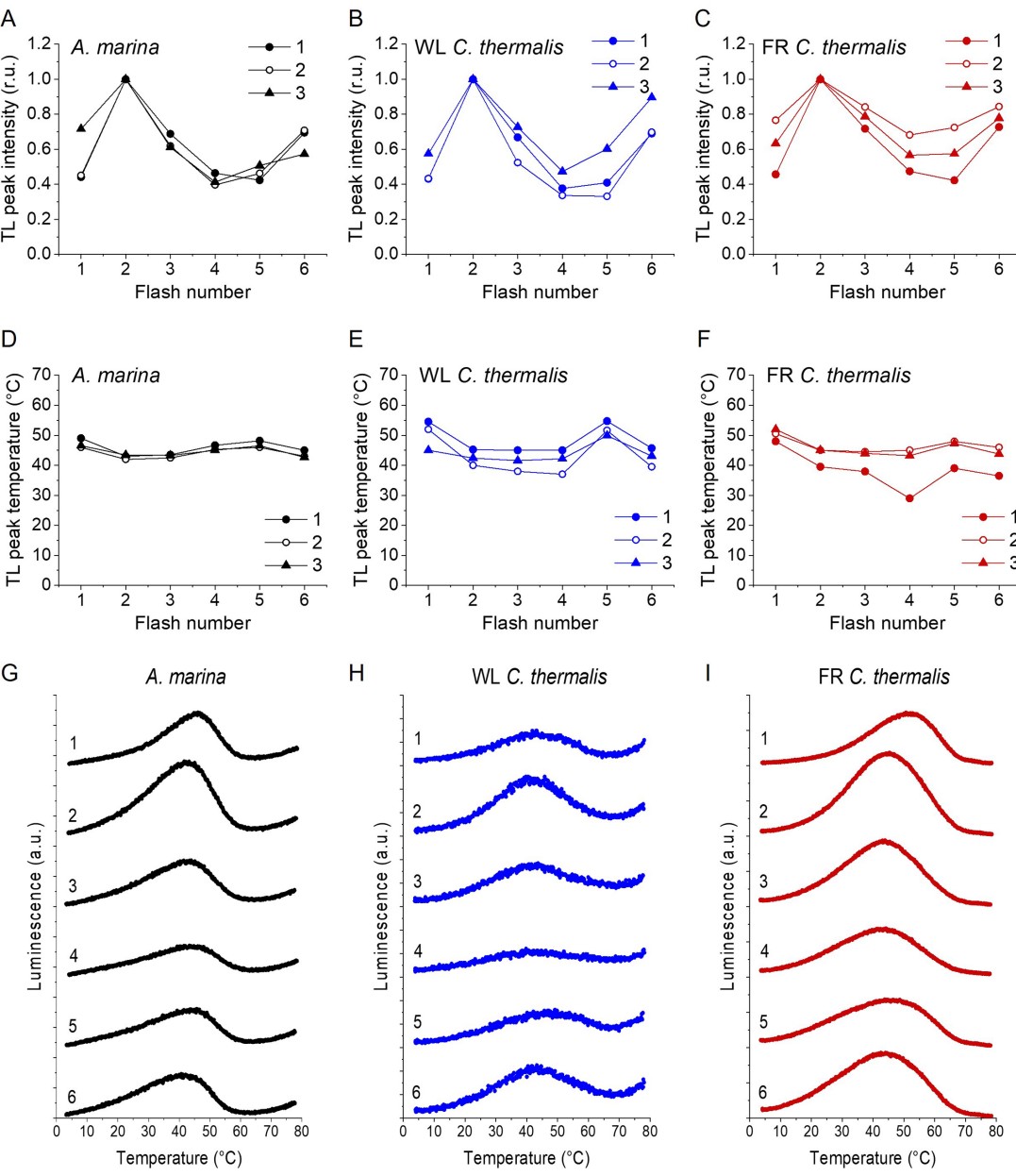

**Appendix 2—figure 1.** Plots of the flash-induced oscillations of the thermoluminescence peak amplitudes (**A, B and C**) and temperatures (**D, E and F**) measured in *A. marina*, WL *C. thermalis* and FR *C. thermalis* membranes. The TL peak amplitudes and temperatures are plotted as a function of the number of flashes given before measuring the thermoluminescence glow curve. Peak amplitudes were normalized to the amplitude value measured after 2 flashes. The membranes, at a final concentration of 5 µg Chl ml⁻¹, were pre-illuminated for ~10 s at room temperature and subsequently dark-adapted on ice for 1 h before the measurements. The flashes were fired at 4 °C at 1 s time intervals, and the samples were then heated from 4°C to 80°C at 1 °C s⁻¹. Each series of data points corresponds to the TL amplitudes and temperatures measured in an independent biological replicate (numbered 1–3, traces in *Appendix 2—figure 1—source data 1*). (**G, H and I**) Representative thermoluminescence glow curves recorded after a train of flashes (from 1 to 6, as indicated by the number next to each curve) in one of the three membrane samples in panels A-F for each strain.

The online version of this article includes the following source data for appendix 2—figure 1:

**Appendix 2—figure 1—source data 1.** Flash dependent thermoluminescence.

## Appendix 3

## Flash-induced S-state turnover measured in the UV

*Appendix 3—figure 1* shows the fit of the flash-induced absorption changes at 291 nm that reflect the progression through the S-states of the Mn-cluster (*Lavorel, 1976*; *Boussac et al., 2004*). The data are those reported in *Figure 4*: absorption changes measured in *T. elongatus* PsbA3-PSII cores with excitation at 680 nm, in *A. marina* membranes with excitation at 680 nm and in partially purified Chl-f-PSII cores from FR *C. thermalis* with excitation at 680 and 750 nm. The measurements were performed in presence of PPBQ, with intervals of 300ms between the flashes.

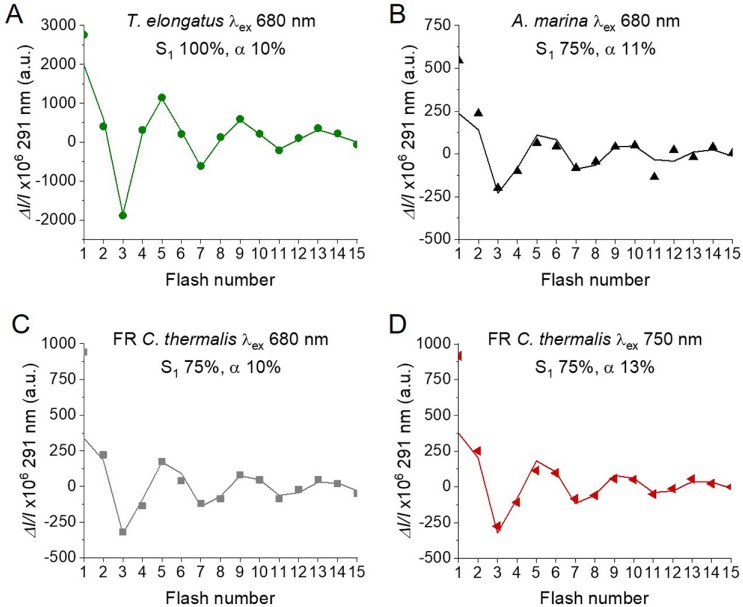

**Appendix 3—figure 1.** Fits of the flash-induced S-state turnover measured as absorption changes at 291 nm in *T. elongatus* PsbA3-PSII cores (**A**), *A. marina* membranes (**B**) and FR *C. thermalis* PSII cores (**C**) with laser excitation at 680 nm and in FR *C. thermalis* PSII cores with laser excitation at 750 nm (**D**). Absorption changes were measured at 100ms after each of a series of saturating flashes fired with a 300 ms time interval. The data are the same as those reported in *Figure 4*, while the lines represent the fits of the experimental data. The initial fraction of PSII in $S_1$ state and the miss factors are indicated (in %).

The fit was done by taking the absorption changes corresponding to the $S_0 \rightarrow S_1$, $S_1 \rightarrow S_2$ and $S_2 \rightarrow S_3$ transitions determined in *T. elongatus* with the procedure established by *Lavergne, 1991*, and multiplying them by a factor γ which corresponds to the ratio in active PSII per chlorophyll of the given sample with respect to the *T. elongatus* sample. It is of note that the factor γ indicates the fraction of active PSII centres over the total PSII present only when comparing isolated cores, while in the data reported here, in which partially purified $O_2$-evolving Chl-f-PSII cores and *A. marina* membranes were used, it merely reflects the amounts of active PSII present in those samples for a given chlorophyll concentration. Since the measurements were done by using single-turnover excitation flashes (6 ns FWHM), the double-hit parameter β was considered to be zero. Using the formula developed by *Lavorel, 1976*, the miss parameter α and the proportion of the centres in $S_1$ state in the dark-adapted samples could be calculated. In these fits, the absorption changes on the first flash of the sequence was not taken into account because they may contain a non-oscillating component (*Lavergne, 1991*). This non-oscillating component is bigger in the *A. marina* and FR *C. thermalis* samples than in the *T. elongatus* sample, likely because of the lower content of active PSII per chlorophyll. The misses were comparable in all samples (~10%), and in the Chl-f-PSII cores from FR *C. thermalis* they did not significantly increase when using 750 nm excitation flashes.

The fits in *Appendix 3—figure 1* indicate that the proportion of centres in $S_1$ in the dark-adapted samples was 75% in the *A. marina* and FR *C. thermalis* samples but 100% in the *T. elongatus* PsbA3-PSII cores. All samples were pre-illuminated in ambient light for ~10 s and then dark-adapted for >1 hr before the measurements and were therefore expected to be in 100% $S_1$ at the start of

the flash sequence, with ~75% of the centres having TyrD$^\bullet$ and ~25% having TyrD (***Styring and Rutherford, 1987***). It has been shown that TyrD can reduce the $S_2$ and $S_3$ oxidation states of the Mn-cluster (***Velthuys and Visser, 1975***): in samples having starting populations of 75% TyrD$^\bullet$S$_1$ and 25% TyrDS$_1$, some of the $S_2$ and $S_3$ states generated during the flash sequence will be re-reduced to $S_1$ and $S_2$, respectively, in centres where TyrD is present. This process will result in the apparent presence of 25% $S_0$ in the dark-adapted sample. This effect has been shown to depend on the spacing between excitation flashes (***Vermaas et al., 1984***): if the time between the flashes is not long enough to allow for TyrD donation, the flash pattern will reflect the initial presence of 100% $S_1$ (34). At room temperature, electron donation from TyrD is slower in *T. elongatus* PSII than in plant PSII (***Sugiura et al., 2004***): this could reflect the fact that *T. elongatus* is a thermophile and mesophilic cyanobacterial species such as *A. marina* and *C. thermalis* could be expected to have TyrD oxidation kinetics more similar to plants, thus explaining the difference in $S_1$ populations in our fits.

## Appendix 4

### Variability of thermoluminescence intensities

*Figure 5—figure supplement 1* shows the plots of the peak amplitudes and temperature of the thermoluminescence arising from $S_2Q_B^-$ and $S_2Q_A^-$ in the three types of PSII. As mentioned in the main text, although our data fit qualitatively with earlier reports (*Nürnberg et al., 2018*; *Cser et al., 2008*), there is a degree of variability in both amplitude and temperature between biological replicates. Consequently, the average values reported in *Table 3* present relatively high standard deviations. The variability in TL intensity between different membrane samples could depend on differences in the $Q_B/Q_B^-$ ratios and distribution of S states present in the dark before applying the single-turnover flash (*Rutherford et al., 1982*).

These variabilities between biological replicates could also partially explain slight discrepancies between the data reported here and those in *Nürnberg et al., 2018* regarding the ratio of luminescence intensity between the Chl-f-PSII and the Chl-a-PSII. In *Nürnberg et al., 2018* the luminescence from both $S_2Q_B^-$ and $S_2Q_A^-$ were reported to be >25 times higher in FR *C. thermalis* membranes than in WL *C. thermalis* membranes, while the data reported here indicate that the luminescence of FR *C. thermalis* is between 5 and 16 times higher than in WL *C. thermalis* in the case of the $S_2Q_B^-$ recombination, and between 3 and 15 times higher in the case of the $S_2Q_A^-$ recombination. In the present work all measurements were performed at constant chlorophyll concentrations (5 µg Chl ml$^{-1}$ in the case of *A. marina* and FR *C. thermalis*, 10 µg ml$^{-1}$ in the case of WL *C. thermalis*), while in Nürnberg et al. the FR *C. thermalis* membranes were diluted to achieve a signal intensity comparable to that obtained in WL *C. thermalis* membranes. Although the dilution factor was included in the normalization on the $O_2$ evolution activities, these differences in the protocols used could contribute to the quantitative discrepancies, together with the biological variability, as at higher chlorophyll concentrations sample self-absorption can occur, thus skewing the measured TL intensity.

## Appendix 5

### Analysis and interpretation of luminescence decay kinetics

The $S_2Q_A^-$ luminescence decay curves measured in *A. marina*, WL *C. thermalis* and FR *C. thermalis* at 10, 20, and 30°C (*Appendix 5—figure 1*) could be fitted with three exponential components (*Appendix 5—table 1*) and the differences in the kinetics between samples and between temperatures could be ascribed to differences in the amplitude and lifetimes of these components.

The luminescence decays at each temperature were similar in shape in Chl-a-PSII and Chl-f-PSII, while they were markedly different in Chl-d-PSII. Chl-a-PSII and Chl-f-PSII had a fast decay phase ($T_1$ ~0.5 and~1 s, respectively) absent in Chl-d-PSII. This phase, that has a bigger amplitude in Chl-a-PSII (~60%) than in Chl-f-PSII (~30%), is too fast to correspond to the $S_2Q_A^-$ recombination and appears to match the rates of $TyrZ^\bullet(H^+)Q_A^-$ recombination occurring either in centres lacking an intact Mn-cluster (*Yerkes et al., 1983*) or in intact centres before charge separation is fully stabilised, as proposed in *Debus et al., 2000*. The contribution of this fast component to the total luminescence emission was no more than 10% in the case of WL *C. thermalis* and 5% for FR *C. thermalis*. This decay phase was not detectable in the case of *A. marina*, suggesting that $TyrZ^\bullet(H^+)Q_A^-$ recombination might be too fast in Chl-d-PSII to appear in our measurements. In *A. marina* an additional slower phase (~40 s) was present at 10 °C, but the very low amplitude made its contribution to the overall decay negligible.

The luminescence decay that we ascribe to the $S_2Q_A^-$ back-reaction in the seconds to tens of seconds timescale, is comprised of two decay components, designated the middle and slow phases in *Appendix 5—table 1*. Both phases were faster in Chl-d-PSII (~3 and~11 s) than in Chl-a-PSII (~4 and~25 s), but slower in Chl-f-PSII (~9 and~39 s). In *A. marina* the lifetimes of these two decay components did not show a significant temperature dependence, resulting in only a minor acceleration of the overall luminescence decay of the Chl-d-PSII between 10°C and 30°C (*Figure 5D*). Indeed, the relative contribution of the two decay phases to the total luminescence changed little in function of temperature in this sample (*Appendix 5—figure 1*), with the changes being only at the level of the amplitude of the decay phases. The middle phase lifetimes did not show a significant temperature dependence in WL and FR *C. thermalis* either, but they were slower than in *A. marina*, especially in FR *C. thermalis*. The slow phase was also slower in the two *C. thermalis* samples and, additionally, its decay accelerated with increasing temperature, while its amplitude decreased. This resulted in its contribution to the overall luminescence decreasing between 10°C and 30°C (*Appendix 5—figure 1*) and the overall decay accelerating significantly (*Figure 5D*), especially in the FR *C. thermalis*.

It can be argued that the differences in kinetics between samples and their changes in function of temperature could represent changes in the relative contribution of different recombination pathways to the decay of $S_2Q_A^-$. It is not clear, though, whether each of the two decay components we identified represents a distinct recombination route or whether they derive from the combination of more complex kinetics. For instance, it has been suggested that the so-called 'deactivation' luminescence should follow a hyperbolic decay, rather than an exponential decay, due to the progressive decrease in the concentration of $S_2Q_A^-$ resulting in a progressive slowing down of the rates of the various recombination routes (*Lavorel and Dennery, 1984*; *Tyystjarvi and Vass, 2007*). The data presented here could be satisfactorily fitted with exponentials but, given the considerations above and the uncertainty about how the evolution of luminescence reflects the actual concentrations of the charge separated states from which it originates, no assignment of the decay phase to specific recombination routes could be made.

Altogether, the data show that the luminescence kinetics in Chl-d-PSII are significantly different from those in Chl-a-PSII and Chl-f-PSII, pointing to a faster decay of the $S_2Q_A^-$ charge separated state.

According to electron tunnelling calculations, the rate of $P_{D1}^+Q_A^-$ direct recombination to ground ($10^2$–$10^3$ $s^{-1}$) is much slower than $P_{D1}^+Phe^-$ recombination to ground ($10^6$–$10^7$ $s^{-1}$), although the limiting rate for $S_2Q_A^-$ recombination via the repopulation of $Phe^-$ is thought to be the migration of the electron hole from the Mn-cluster to TyrZ (~$10^3$ $s^{-1}$) (*Sugiura et al., 2014*; *Moser et al., 2005*). Although the temperature dependence of the recombination routes is complex, an increase in temperature would have no effect on the rate of $P_{D1}^+Q_A^-$ direct recombination to ground, but would increase the rate of the backwards electron transfer from $Q_A^-$ to Phe, as this is thermally activated, following the relationship

$$k_{rev} = k_{fwd} \cdot e^{-\frac{\Delta G^0}{k_B T}} \tag{1}$$

where $k_{rev}$ and $k_{for}$ are the rate constants of the backward and forward electron transfer, respectively, ΔG is the energy gap between the two cofactors, $k_B$ is the Boltzmann constant and T is the temperature. Note that the rates of back-transfer of the positive charge from the Mn-cluster to $P_{D1}$ are also thermally activated and thus will accelerate with temperature and affect the rates of recombination from Phe⁻. In this case, though, the smaller ΔGs involved should result in a less pronounced temperature dependence compared to the back electron transfer from $Q_A^-$ to Phe (according to the equation above).

The acceleration of the luminescence decay kinetics with increasing temperature, observed for Chl-a-PSII and Chl-f-PSII, could reflect an increase in the contribution of $S_2Q_A^-$ recombination route via repopulation of Phe⁻ in competition with the direct, non-radiative $P_{D1}^+Q_A^-$ recombination route. In Chl-d-PSII, the lower temperature of the $S_2Q_A^-$ recombination thermoluminescence peak (*Figure 5B* and *Figure 5—figure supplement 1*) suggests a smaller ΔG between $Q_A$ and Phe. This would result in a faster electron transfer from $Q_A^-$ back to Phe, with this route already dominating the competition with the direct $P_{D1}^+Q_A^-$ recombination to ground, with a consequently high luminescence yield and a small temperature sensitivity of the decay rates.

**Appendix 5—table 1.** Time constants and relative amplitudes of the different phases of luminescence decay obtained by fitting the data recorded at 10, 20, and 30°C with a three-exponential equation.

The values represent the averages of 3 biological replicates,± s.d. The fast decay phase is assigned to TyrZ•(H⁺)$Q_A^-$ recombination, while the middle and slow phases are assigned to $S_2Q_A^-$ recombination. The additional phase identified in *A. marina* membranes at 10 °C is unassigned.

| Strain and temperature | Fast phase T$_1$/Amp (s/%) | Middle phase T$_2$/Amp (s/%) | Slow phase T$_3$/Amp (s/%) | Additional phase T$_4$/Amp (s/%) |
|---|---|---|---|---|
| *A. marina* | | | | |
| 10 °C | -/- | 3.5±1.0 / 64±22 | 10.6±2.9 / 35±18 | 36.5±7.3 / 5±3 |
| 20 °C | -/- | 3.2±0.6 / 88±5 | 12.7±2.9 / 18±3 | -/- |
| 30 °C | -/- | 3.0±0.4 / 87±9 | 10.2±2.4 / 19±7 | -/- |
| WL *C. thermalis* | | | | |
| 10 °C | 0.5±0.2 / 72±8 | 4.0±2.4 / 18±4 | 32.0±6.8 / 7±2 | -/- |
| 20 °C | 0.4±0.1 / 62±13 | 3.2±1.2 / 23±7 | 19.1±2.8 / 12±5 | -/- |
| 30 °C | 0.6±0.1 / 55±7 | 6.0±0.8 / 32±4 | 16.9# / 9# | -/- |
| FR *C. thermalis* | | | | |
| 10 °C | 1.0±0.2 / 43±2 | 10.4±1.0 / 26±2 | 43.7±1.8 / 31±2 | -/- |
| 20 °C | 1.0±0.1 / 28±4 | 7.9±0.7 / 35±3 | 23.5±1.8 / 38±4 | -/- |
| 30 °C | 1.4±0.4 / 14±10 | 8.6±1.1 / 72±18 | 18.6±3.7 / 15±8 | -/- |

#The slow phase in WL *C. thermalis* membranes at 30 °C could be reliably fitted only in one replicate out of three.

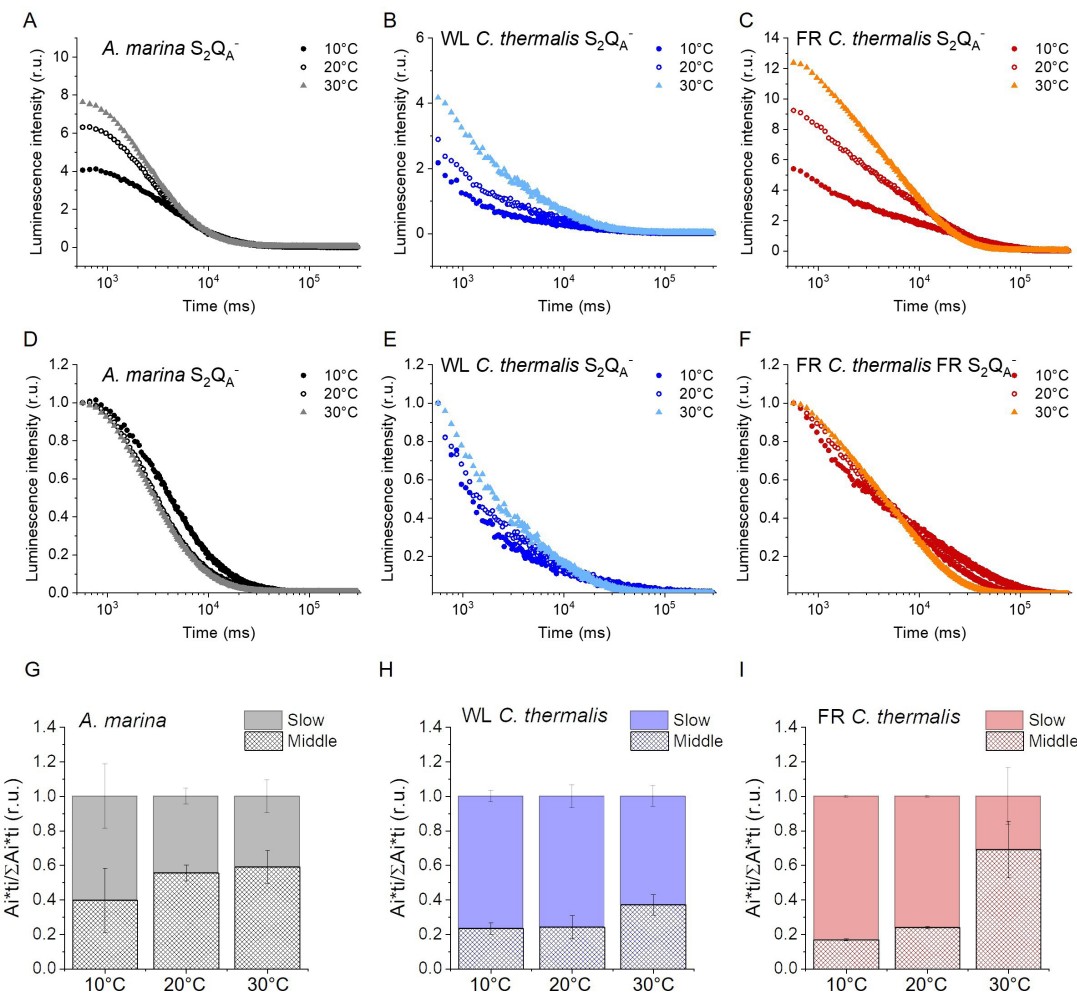

**Appendix 5—figure 1.** Luminescence decay kinetics. (**A, B and C**) Representative $S_2Q_A^-$ luminescence decay curves measured in *A. marina*, WL *C. thermalis* and FR *C. thermalis* membranes in the presence of DCMU. The measurements were performed at 10, 20°C and 30°C. The luminescence decays were measured for 300 s after the flash, and the time is plotted on a logarithmic scale. The luminescence intensities are normalized on the content of $O_2$-evolving PSII of each sample, measured as the maximal oxygen evolution rates under saturating illumination. (**D, E and F**) The same curves as in (A-C) after normalization on the initial intensities. (**G, H and I**) Relative contributions of the middle and slow decay components to the total luminescence emission arising from $S_2Q_A^-$ recombination, calculated using the values in *Appendix 5—table 1*. All luminescence traces are provided in *Appendix 5—figure 1—source data 1*.

The online version of this article includes the following source data for appendix 5—figure 1:

**Appendix 5—figure 1—source data 1.** Luminescence decay kinetics.

## Appendix 6

## Singlet oxygen production and sensitivity to high light in the far-red PSII

### Light sources

Given the different pigments involved in light capture in the three types of PSII studied here, the comparability of experiments could be adversely affected by differences in excitation rates due to the degree of matching of the absorption spectrum of the PSII with excitation spectrum of the light sources used. *Appendix 6—figure 1A* shows the absorption spectra of the three membrane preparations, containing the 3 types of PSII, and the spectral profiles of the xenon lamp and the 660 nm LED. For both light sources the WL and the FR *C. thermalis* samples have a greater spectral overlap with the actinic light spectrum, than does the *A. marina* sample. It can be concluded that under identical illumination conditions, *A. marina* membranes would receive fewer photons during a period of illumination compared to two *C. thermalis* samples.

*Appendix 6—figure 1B* shows the oxygen evolution in the presence of the electron acceptor system, and oxygen consumption rates in the presence of DCMU and histidine measured in *A. marina* membranes as a function of the light intensity. The figure shows experiments done in three biological replicates. Both rates showed a comparable dependence on light intensity and saturated at 7100 µmol photons $m^{-2} s^{-1}$, the intensity used in all the oxygen measurements. The same light intensity was saturating also in the case of WL and FR *C. thermalis* membranes used at the same concentration of 5 µg Chl $ml^{-1}$ (*Appendix 6—figure 1C*).

*Appendix 6—figure 1D and E* show that both the LED and the xenon lamp gave the same rates of $O_2$ evolution, and given their different actinic spectra, this indicates that both were saturating under the conditions of the experiment.

### Singlet oxygen production experiments: the presence of the Mn-cluster

To test whether the $^1O_2$ production in *A. marina* was related to the fraction of PSII centres devoid of an intact Mn-cluster, which is the most obvious functional difference between *A. marina* membrane samples and those from the WL and FR *C. thermalis*, we compared $^1O_2$ formation in untreated and Tris-washed membranes. Tris-washing was used to remove the Mn-cluster from all PSII. As shown in *Appendix 6—figure 2*, the Tris-washed membranes did not display any $O_2$ evolution activity in presence of the acceptors DCBQ and potassium ferricyanide but retained the same $^1O_2$ production capacity as the untreated sample. This indicates that $^1O_2$ formation in *A. marina* is not related to the fraction of centres that are capable of water oxidation.

### Singlet oxygen production: does PSI contribute to $O_2$ uptake?

We tested whether light-induced oxygen consumption observed in *A. marina* membranes could be derived from Photosystem I (PSI) turnover. It is well-known that PSI can reduce $O_2$ to $O_2^{\bullet-}$ and this is greatly enhanced by methyl viologen (MV) acting as a redox mediator. The PSI electron donors, plastocyanin or cytochrome $c_6$, which are both soluble in the lumen, are expected to be lost during preparation of the membranes. As a result, illumination is likely to accumulate oxidized $P_{700}$ resulting in PSI being non-functional. To confirm this in *A. marina* membranes in which PSII activity was blocked by DCMU, we tested whether methyl viologen could induce a light-dependent oxygen consumption in the absence of the histidine $^1O_2$ trap. In isolated *A. marina* membranes, no MV-mediated oxygen consumption was observed in presence of DCMU unless the exogenous PSI electron donors, ascorbate and TMPD, were also added (*Appendix 6—figure 3*). This demonstrates that under the conditions of the experiments used to estimate $^1O_2$ trapping by histidine in the isolated *A. marina* membranes, there was no contribution from PSI activity.

### Singlet oxygen production in cells

We tested if differences in the stability of the membrane samples could explain the marked increase in singlet oxygen production in *A. marina* compared to both the WL and FR the *C. thermalis* samples (see *Figure 6* and related text). Lower stability of PSII in the isolated membranes of *A. marina* was suggested by the presence of long-lived non-decaying emission observed when measuring fluorescence decay kinetics (*Figure 2*) and attributed to a fraction of centers devoid of an intact Mn-cluster. Additionally, the presence of "free" chlorophyll (not excitonically coupled to a photosynthetic complex) in isolated membranes could also lead to singlet oxygen production. We therefore used the histidine trapping method to compare the rates of singlet oxygen production in *A. marina* and FR *C. thermalis* intact cells. The reliability of the His-

**Appendix 6—table 1.** Rates of oxygen evolution (in presence of the electron acceptors DCBQ and ferricyanide) and consumption (in presence of DCMU and DCMU +histidine) measured in *A. marina*, WL *C. thermalis* and FR *C. thermalis* membranes and intact cells under saturating illumination (xenon lamp, 7100 µmol photons $m^{-2}$ $s^{-1}$).

The reported rates are those used to make *Figure 6* and *Appendix 6—figure 4*.

$\mu mol\ O_2\ h^{-1}\ mg\ Chl^{-1}$

| | Membranes | | | Cells | | |
|---|---|---|---|---|---|---|
| *A. marina* | Acceptors | DCMU | DCMU +His | Acceptors | DCMU | DCMU +His |
| Replicate 1 | 114 | 2 | −118 | 214 | −11 | −102 |
| Replicate 2 | 99 | −40 | −93 | 207 | -9 | −232 |
| Replicate 3 | 259 | | −203 | 255 | −10 | −148 |
| Replicate 4 | 205 | | −188 | | | |
| Replicate 5 | 288 | | −147 | | | |
| Replicate 6 | 163 | 0 | −124 | | | |
| WL *C. thermalis* | Acceptors | DCMU | DCMU +His | Acceptors | DCMU | DCMU +His |
| Replicate 1 | 183 | −15 | −20 | 177 | 24 | −15 |
| Replicate 2 | 52 | −22 | −16 | | | |
| Replicate 3 | 69 | -3 | −32 | | | |
| FR *C. thermalis* | Acceptors | DCMU | DCMU +His | Acceptors | DCMU | DCMU +His |
| Replicate 1 | 199 | 10 | −20 | 80 | 4 | −10 |
| Replicate 2 | 221 | −26 | −18 | 193 | 14 | −40 |
| Replicate 3 | 65 | | -6 | 225 | −12 | −83 |
| Replicate 4 | 140 | | −63 | | | |
| Replicate 5 | 57 | 7 | −28 | | | |
| Replicate 6 | 140 | -2 | −17 | | | |

trapping method to monitor $^1O_2$ production in intact cyanobacterial cells has been previously demonstrated (*Rehman et al., 2013*).

*Appendix 6—figure 4A* shows that in *A. marina* cells the rate of histidine-mediated oxygen uptake was much higher, relative to the maximal oxygen evolution rate, than in FR *C. thermalis*. The values obtained in cells were comparable with those obtained in isolated membranes, despite variability between biological replicates (this variability makes the difference between the two strains less significant than that measured in membranes, *P*=0.08). Like FR *C. thermalis* cells, WL *C. thermalis* cells also showed low levels of $^1O_2$ production (*Appendix 6—figure 1B*), similar to those measured in the respective membranes. It is of note that both in membranes and intact cells, the rates of maximal $O_2$ evolution (measured in presence of exogenous electron acceptors) and of $^1O_2$ production (measured in presence of DCMU) do not depend on the functionality of the electron transport chain downstream of PSII.

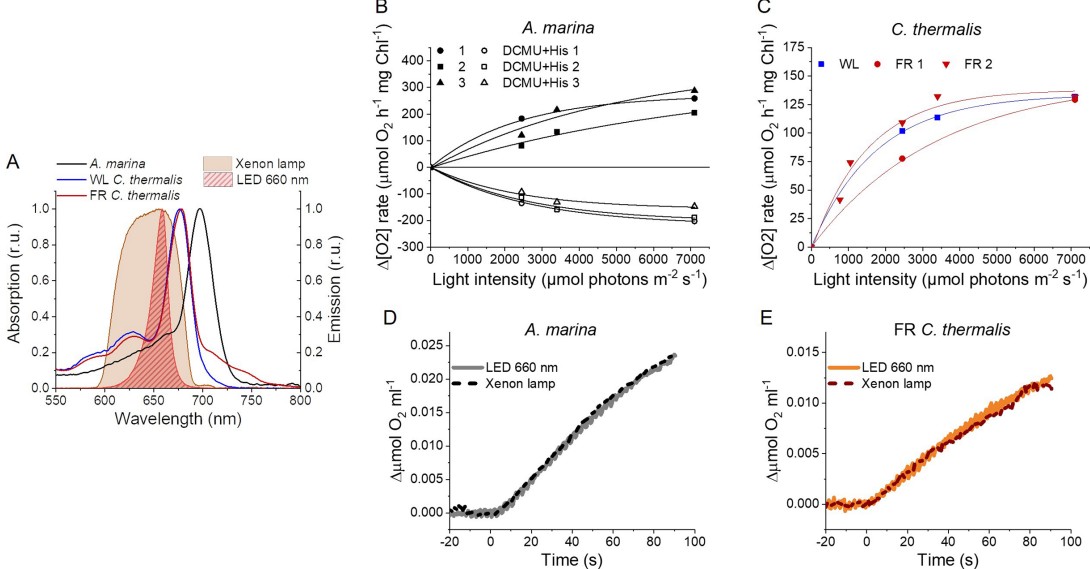

**Appendix 6—figure 1.** Light sources used for $^1O_2$ production measurements and high light treatment.
(**A**) Absorption spectra (normalized on the maximal absorption in the Qy region) of *A. marina*, WL *C. thermalis* and FR *C. thermalis* membranes and spectral profiles (normalized on the maximal emission) of the 660 nm LED and xenon lamp used. (**B**) Light saturation curves of $O_2$ evolution (in presence of DCBQ and ferricyanide, solid symbols) and $^1O_2$ production (in the presence of DCMU and histidine, open symbols) in three biological replicates of *A. marina* membranes, using the xenon lamp. The intensity of the lamp was decreased by using neutral filters. (**C**) Light saturation curves of $O_2$ evolution in WL *C. thermalis* (1 biological replicate) and FR *C. thermalis* (2 biological replicates) membranes, used at a final Chl concentration of 5 µg ml$^{-1}$. (**D and E**) Representative $O_2$ electrode traces (shown after subtraction of the dark baseline) monitoring maximal $O_2$ evolution in *A. marina* and FR *C. thermalis* membranes, used at a final Chl concentration of 5 µg ml$^{-1}$. Measurements were performed in presence of DCBQ and potassium ferricyanide using either the 660 nm LED (2600 µmol photons m$^{-2}$ s$^{-1}$) or the xenon lamp (7100 µmol photons m$^{-2}$ s$^{-1}$) for illumination. All data are provided in ***Appendix 6—figure 1—source data 1***.

The online version of this article includes the following source data for appendix 6—figure 1:

**Appendix 6—figure 1—source data 1.** Light sources and saturation curves.

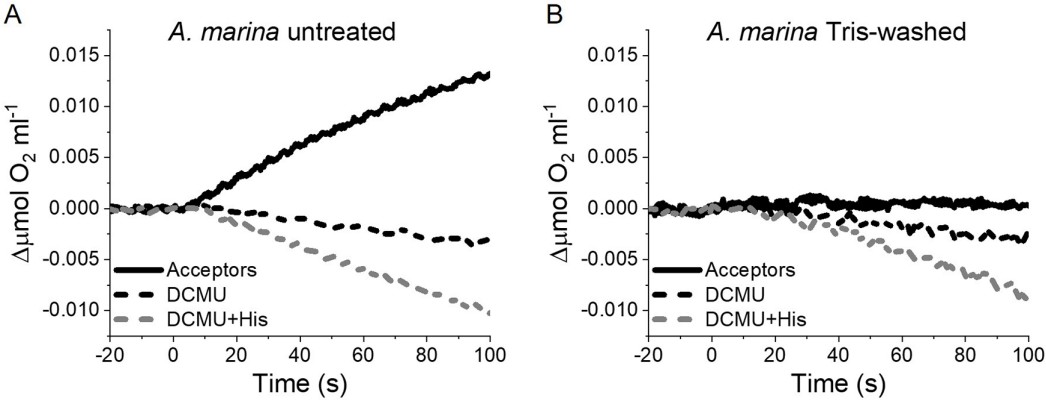

**Appendix 6—figure 2.** $^1O_2$ formation in presence of DCMU measured as the rate of histidine-dependent consumption of $O_2$ induced by saturating illumination in untreated (**A**) and Tris-washed (**B**) *A. marina* membranes. Measurements were performed in the presence of DCBQ and potassium ferricyanide (Acceptors) or in presence of DCMU, with or without the addition of L-Histidine (His). All traces are shown after subtraction of the dark baseline (traces in ***Appendix 6—figure 2—source data 1***).

The online version of this article includes the following source data for appendix 6—figure 2:

**Appendix 6—figure 2—source data 1.** Tris-washed *A. marina* singlet oxygen.

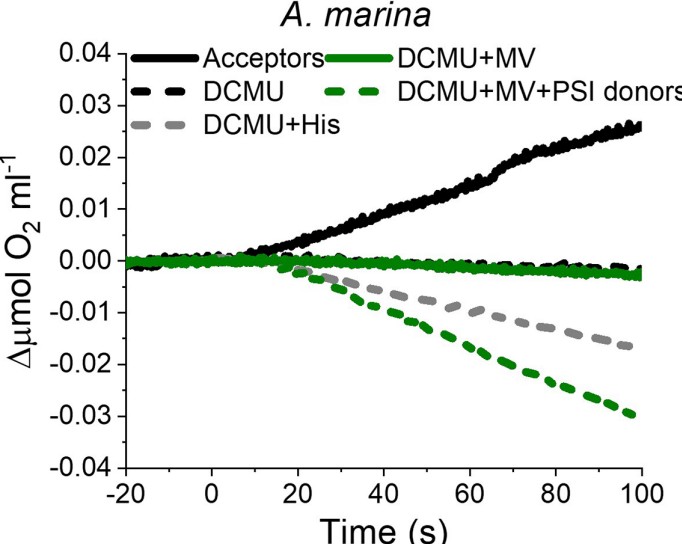

**Appendix 6—figure 3.** $O_2$ electrode traces (shown after subtraction of the dark baseline) monitoring $O_2$ evolution and uptake in *A. marina* membranes; $^1O_2$ formation is monitored by $O_2$-uptake due to $^1O_2$ scavenging by histidine. Measurements were performed in the presence of DCBQ and potassium ferricyanide (Acceptors) or in the presence of DCMU, with or without the addition of L-Histidine (His). PSI activity (green traces) was measured as the rate of methyl viologen (MV, 100 µM)-dependent oxygen consumption in the presence of DCMU, either with (dashed green line) or without (solid green line, 'PSI donors') the electron donors ascorbate (5 mM) and TMPD (50 µM). Traces are provided in ***Appendix 6—figure 3—source data 1***.

The online version of this article includes the following source data for appendix 6—figure 3:

**Appendix 6—figure 3—source data 1.** A. *marina* PSI activity.

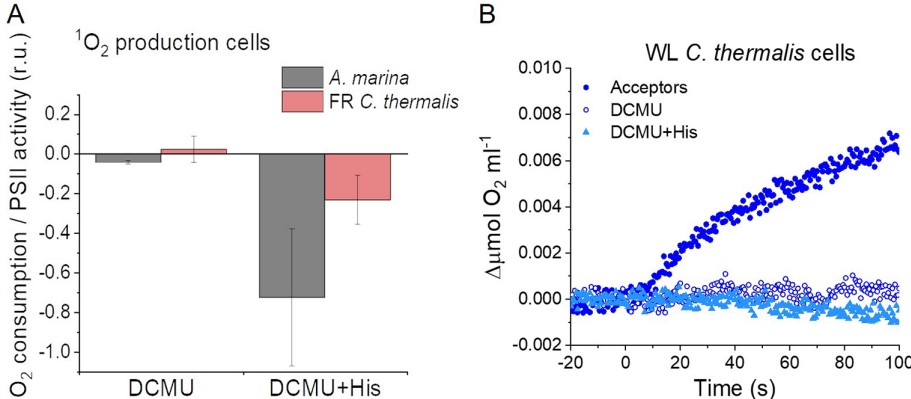

**Appendix 6—figure 4.** Singlet oxygen production in intact cells. (**A**) $^1O_2$ formation in presence of DCMU measured as the rate of histidine-dependent consumption of $O_2$ induced by saturating illumination in *A. marina* and FR *C. thermalis* cells. The data are averages (±s.d.) of 3 biological replicates for each strain. For each replicate, the rates of oxygen consumption were normalized to the maximal oxygen evolution rates obtained with the same illumination in the presence of the exogenous acceptors, DCBQ and ferricyanide. The non-normalized rates of each replicate are provided in ***Appendix 6—table 1***. (**B**) $O_2$ electrode traces (shown after subtraction of the dark baseline) monitoring $O_2$ evolution and uptake in WL *C. thermalis* cells. $^1O_2$ production in the presence of DCMU was measured as the rate of histidine-dependent consumption of $O_2$ induced by saturating illumination (xenon lamp, 7100 µmol photons $m^{-2} s^{-1}$). Measurements were performed in the presence of DCBQ and ferricyanide (Acceptors), or in the presence of DCMU, with or without the addition of histidine (His). All traces are provided in ***Appendix 6—figure 4—source data 1***.

The online version of this article includes the following source data for appendix 6—figure 4:

**Appendix 6—figure 4—source data 1.** Singlet oxygen production cells.

## Appendix 7

## Supplementary materials and methods

### Isolation of membranes

Cells were harvested by centrifugation at 6,000 x g for 5 min and resuspended in ice-cold buffer (50 mM MES-NaOH pH 6.5, 5 mM CaCl$_2$, 10 mM MgCl$_2$, 1.2 M betaine and 20% v/v glycerol) with a protease inhibitor mixture (1 mM aminocaproic acid, 1 mM benzamidine, and 0.2% (w/v) bovine serum albumin) and 0.5 mg ml$^{-1}$ DNaseI. All following steps were performed on ice under dim green light. *A. marina* and *C. thermalis* cells were broken by two passages through a cell disruptor (Constant System, Model T5) at a pressure of 25 kPsi. *Synechocystis* cells were broken with bursts of vortexing with glass beads. Unbroken cells were removed by centrifugation for 5 min at 1000 x g, 4 °C. Membranes were pelleted by centrifugation at 125,000 x g and 4 °C for 30 min and washed three times with resuspension buffer. Membranes were resuspended in resuspension buffer, frozen in liquid nitrogen and stored at –80 °C.

### Removal of Mn-cluster by Tris-washing of membranes

*A. marina* membranes were diluted in ice-cold 1 M Tris pH 9.5 plus 3 mM EDTA to a final chlorophyll concentration of 190 µg ml$^{-1}$ and incubated on ice under ambient light with continuous stirring for 30 min at 4 °C. The membranes were then washed twice in ice-cold resuspension buffer (the same used for membrane isolation) and finally resuspended in the same.

### Analysis of Q$_A^-$ reoxidation kinetics as measured by fluorescence

The flash-induced chlorophyll fluorescence curves were fitted with a linear combination of two exponentials (fast and middle phase) and a hyperbolic component (slow phase), where Ft is the variable fluorescence yield, F$_0$ is the basic fluorescence level before the flash, A$_1$–A$_3$ are the amplitudes and T$_1$–T$_3$ are the time-constants, based on *Crofts and Wraight, 1983*; *Vass et al., 1999*.

$$F_t - F_o = A_1 \cdot \exp\left(-t/T_1\right) + A_2 \cdot \exp\left(-t/T_2\right) + A_3/\left(1 + t/T_3\right) \tag{1}$$

In order to better resolve the µs to ms components associated with forward electron transfer from Q$_A^-$ to Q$_B$ or Q$_B^-$, the same curves but truncated at 1 s were fitted using a three exponentials decay and am off-set (y$_0$) accounting for the non-decaying signal in the time-window:

$$F_t - F_o = A_1 \cdot \exp\left(-t/T_1\right) + A_2 \cdot \exp\left(-t/T_2\right) + A_3 \cdot \exp\left(-t/T_3\right) + y_0 \tag{2}$$

The curves obtained in presence of 20 µM DCMU (3-(3,4-dichlorophenyl)–1,1-dimethylurea) could be fitted with two phases (one exponential and one hyperbolic) for *A. marina* and three phases (two exponentials and one hyperbolic) for WL and FR *C. thermalis*, because of the presence in both types of *C. thermalis* samples of a small initial fast phase, which probably corresponds to a small fraction of PSII centers where DCMU did not bind, as previously suggested (*Cser et al., 2008*).

### Thermoluminescence and luminescence

For the S$_2$Q$_A^-$ and S$_2$Q$_B^-$ TL measurements, samples were cooled to –20 °C and excited with a single turnover saturating laser flash (Continuum Minilite II, frequency doubled to 532 nm, 5 ns FWHM). The samples were then incubated in the dark at –20 °C for 30 s, before heating from –20°C to 80°C at 1 °C s$^{-1}$. The amplitudes of the TL peaks were normalized on the basis of the maximal oxygen evolution rates measured for each sample. For the measurement of the flash-dependence of TL, the samples were cooled to 4 °C and excited with a single or multiple saturating laser flashes fired at 1 s time intervals. Samples were then heated from 4°C to 80°C at 1 °C s$^{-1}$.

S$_2$Q$_A^-$ luminescence decay measurements were performed at a constant (ΔT<0.2 °C) temperature of either 10, 20 or 30 °C in presence of 20 µM DCMU. The samples were pre-equilibrated for 10 s in darkness at the given temperature before being excited with a single turnover saturating laser flash. Luminescence was then recorded from 570ms to 300 s after the flash. The total luminescence emission was calculated as the integrated area below the decay curves normalized on the basis of the maximal oxygen evolution rates measured for each sample. The measured curves were fitted with a linear combination of three exponential components where L is the luminescence, A$_1$–A$_3$ are the amplitudes and T$_1$–T$_3$ are the lifetimes.

$$L(t) = A_1 \cdot \exp\left(-t/T_1\right) + A_2 \cdot \exp\left(-t/T_2\right) + A_3 \cdot \exp\left(-t/T_3\right) \tag{3}$$

The average decay lifetime was calculated from the exponential components 2 and 3 as follows:

$$\tau_{av} = \sum_i A_i T_i / \sum_i A_i \tag{4}$$

The contribution of each luminescence decay component to the total luminescence emission was calculated as

$$L_i = A_i T_i / \sum_i A_i \cdot T_i \tag{5}$$

## UV transient absorption

In the UV pump-probe absorption measurements performed using a lab-built Optical Parametric Oscillator (OPO)-based spectrophotometer, the single-turnover excitation flashes were provided by a Nd:YAG laser (Surelite II, Amplitude Technologies) at 532 nm, which pumped an OPO (Surelite OPO plus) producing monochromatic saturating flashes (6 ns FWHM) at the indicated wavelengths. The power of the flashes at the wavelengths used, measured at the level of the laser output, was: 2.7 mJ at 680 nm, 2.7 mJ at 720 nm, 3.8 mJ at 727 nm, 3.3 mJ at 734 nm, 3.7 mJ at 737 nm, 4 mJ at 749 mJ. The optics components between the laser output and the cuvettes containing the sample induce the same attenuation at all wavelengths. When indicated, the flash intensity was attenuated by 17% using a metal grid. Detecting flashes were provided by an OPO (Horizon OPO, Amplitude Technologies) pumped by a frequency tripled Nd:YAG laser (Surelite II, Amplitude Technologies), producing monochromatic flashes (291 nm, 2 nm full-width at half-maximum) with a duration of 5 ns. The time delay between the laser delivering the excitation flashes and the laser delivering the detecting flashes was controlled by a digital delay/pulse generator (DG645, Stanford Research). The light-detecting photodiodes were protected from transmitted and scattered actinic light and fluorescence by BG39 Schott (Mainz, Germany) filters.

## Flash-dependent oxygen evolution with Joliot electrode

For each measurement, membranes equivalent to 10 μg of total chlorophyll, brought to 750 μl with buffer A (150 mM NaCl, 25 mM MES, 1 M glycine betaine, 5 mM $MgCl_2$, and 5 mM $CaCl_2$, pH 6.2) were deposited on the electrode assembly, which was then centrifuged in a swing-out rotor at 10,000×g for 10 min (at 4 °C). Using a home-built potentiostat, which provided an electrode polarization of –0.95 V (switched on 15 s before the first excitation flash), the current signal was recorded 20ms before and 480ms after each light flash, for a total of 40 flashes with a flash-spacing of 900 ms. The current signal reflects the $O_2$ reduction process at the bare platinum electrode. Three different light sources were used to induce the S-state transitions: a custom-made LED flashing device with two changeable high-performance LEDs (Luminus) and a Xenon flashlamp (EG&G Optoelectronics). The LEDs had emission peaks in the red and far-red (613 nm and 730 nm respectively) and the flashlamp was equipped with 570 nm cut-off filter suppressing shorter wavelengths and thereby photoelectric artefacts. The total energy per light flash was determined with a 1 $cm^2$ power meter (Ophir Photonics) at the exit of the light guide, which conveyed the light to the sample. The energy of the LED flashes (40 μs FWHM) was 270 μJ for the red LED and 210 μJ for the far-red LED, whereas for the flashlamp pulse (10 μs FWHM) it was 540 μJ. During the data acquisition the sample was kept at 20 °C using a Peltier and monitored by a temperature sensor immersed in the sample buffer.

