## [Editor Report]

This manuscript describes the energetic mechanisms by which two quite different cyanobacteria use far-red light. The work describes the energetic constraints and preferred operating conditions of these "strategies" in particular on how nature has solved the problem of low energy "headroom'" required to prevent deleterious back reactions while maintaining efficient energy storage. The differences between the species are quite interesting and show that nature has evolved multiple solutions to fundamental limitations. Given the importance of understanding and improving the efficiency of photosynthesis, and the new insights revealed, the work will be of interest to a broad audience.

---

## [Decision Letter]

**Decision letter after peer review:**

Thank you for submitting your article "Impact of energy limitations on function and resilience in long-wavelength Photosystem II" for consideration by *eLife*. Your article has been reviewed by 4 peer reviewers, and the evaluation has been overseen by a Reviewing Editor and Benoît Kornmann as the Senior Editor. The following individuals involved in the review of your submission have agreed to reveal their identity: Jingcheng Huang (Reviewer #1); Elisabet Romero (Reviewer #2).

Essential revisions:

This manuscript describes the energetic mechanisms by which two quite different cyanobacteria use far-red light.

Four excellent and thoughtful reviews were obtained. They all expressed that the experiments and data analysis were carefully done and generally support their conclusions. However, some important issues were brought up that should be addressed.

Depending on the specific replies, it will most likely be possible to address these without further experimentation, but some additional work may be required to rule out some alternative explanations.

See in particular comments on the PQ pool and S-states status and misses, and distinguishing singlet oxygen produced by reaction centers and antenna complexes, by Reviewers 1 and 3. Addressing the other (quite constructive) comments should also help clarify and increase the impact of the text.

*Reviewer #2 (Recommendations for the authors):*

The authors present a comprehensive investigation of the photochemical efficiency and resilience to photodamage of two low-energy Chl-d and Chl-f PSII systems by applying a collection of carefully designed and executed spectroscopic and electrochemical methods. The data presentation in the figures and the explanations provided in the text are clear and concise, and the large number of experimental biological replicas measured together with the comparison with the relevant controls ensure the consistency of the data presented. The conclusions reached are fully supported by the data presented in the main text and further confirmed by the large dataset presented in the supplementary information. Excellent work, in my opinion.

Overall, I consider that this work is methodologically and scientifically rigorous, objectively presented, and of broad interest, and therefore it fulfills the high-quality standards of *eLife*. For all these reasons I recommend this manuscript for publication in *eLife*.

*Reviewer #3 (Recommendations for the authors):*

This is a generally solid manuscript and the contents will be of great value to the photosynthesis research community, as discussed in my public review. However, there are a few points which should be addressed:

1) Have all the authors reviewed this? Stefano Santabarbara's name appears to be misspelled.

2) Particularly in the Introduction, this could use a quick review for spelling and grammar.

3) The abbreviation for pheophytin is given as both "Phe" and "Pheo".

4) Line 102: "the locations of the 4 antenna in the peripheral antenna" – I believe what there are 4 of here would be Chl-f.

5) Lines 117-118: The decision to remove phycobilisomes is understandable and even necessary considering the techniques used, but undercuts the ecological/agricultural interpretations of the results to some extent. Modification of the antenna alongside FRL adaptation is normal, and the lack of this component removes the potential regulating factor of another energy funnel. It might be worth mentioning in the Discussion.

6) Throughout this work, charge recombination is treated as being entirely with the S2 state. Is there any evidence that this is the case? Providing initial S-state populations as obtained from the Kok model used to generate misses might help here. I don't consider this terribly necessary for the main text but it would probably be helpful SI; regardless, all recombination observed is probably with a mixed population of mostly S2 that would be difficult to further clarify but how much of this is actual S2 would be good to know.

7) Line 180-181: It's mentioned that there was not enough PSII in WL membranes from C. thermalis to use as a control for oximetry- this is a bit concerning considering there was apparently enough to generate singlet oxygen. Was using a higher density of membranes not an option? Or, for that matter, additional purification to cores as was done later?

8) What are the flash yields in Figure 3 normalized to? Is it possible to present this data in oxygen per chlorophyll or oxygen per PSII, since it appears the chlorophyll concentration would be known?

9) Line 211: What in C. thermalis membranes makes them scatter so much more light than other strains studied?

10) Line 214-215: For the sake of novice readers, it may be useful to clarify that PsbA3 has the highest sequence identity with Chl-f-PSII D1 out of the T. elongatus PsbA isoforms rather than out of known D1 sequences.

11) Line 218: Please define deltaI/I for first use, this is a fairly niche term.

12) Figure 4: Why is there no large non-oscillating component in T. elongatus absorption and did its absence have any impact on how data modeling was performed?

13) The subsection numbering skips from 3.2 to 3.5.

14) Figure 7 might be better off higher up – it's three pages below its first mention.

15) Line 586: What is the intensity and source of room light used?

16) Were any electron acceptors used in the flash oxygen evolution experiments? On one hand, this is directly discussed in the introduction as necessary to avoid unwanted regulation by the redox poise of the pool in whole cells; on the other, a bare platinum electrode is used.

17) Why was the Kok model described in reference 28 used? This is a fairly outdated model; its creator has developed more detailed ones in the meantime, as have several other groups. It might be worthwhile to reprocess with a newer method, especially when considering the issue of the initial S state distribution not being given.

18) Table S1: With no addition, the slow phase is provided with an amplitude but no time constant, which shouldn't be possible with the model provided- is that just y0 in place of the amplitude?

19) SI lines 163-170: this appears to say that first, Synechocystis membranes have faster fluorescence decay kinetics than intact cells, and later, that Synechocystis has faster decay kinetics in cells than in membranes. Please clarify.

20) Figure S6: The caption references dashed lines; there are no dashed lines in the figure.

*Reviewer #4 (Recommendations for the authors):*

Importantly, I encourage the authors to remove the ambiguity regarding which Chl type is found in the ChlD1 site of Chl-f-PSII. I find that the evidence for Chl d occupying this site in Synechococcus sp. PCC 7335 is overwhelming, and every aspect of that site is well conserved in C. thermalis and all other FaRLiP-capable cyanobacteria. If there is some argument against this sentiment, the authors could include it. For example, if the authors think the two organisms are different in which pigment is found in ChlD1, it could be mentioned.

---

## [Author Response]

Essential revisions:This manuscript describes the energetic mechanisms by which two quite different cyanobacteria use far-red light.Four excellent and thoughtful reviews were obtained. They all expressed that the experiments and data analysis were carefully done and generally support their conclusions. However, some important issues were brought up that should be addressed.Depending on the specific replies, it will most likely be possible to address these without further experimentation, but some additional work may be required to rule out some alternative explanations.See in particular comments on the PQ pool and S-states status and misses, and distinguishing singlet oxygen produced by reaction centers and antenna complexes, by Reviewers 1 and 3. Addressing the other (quite constructive) comments should also help clarify and increase the impact of the text.

We thank the reviewers and the editor for the thoughtful comments and suggestions. We have tried to address them all in the best possible way. The point-by-point answers are provided below, and the revised manuscript with changes highlighted in red is provided in the Related manuscript file.

Additionally, we have rearranged the format of the manuscript, to comply with *eLife*’s policy of incorporating as much information as possible within the main text of the article and to replace a single Supplementary Information file with either supplementary figures linked to main figures or appendices, where additional text is required. More specifically:

– Figure S1 (100 s fluorescence decay kinetics) is now provided as Figure 2—figure supplement 1, Tables S1 and S2 have been merged in Table 1 (all fluorescence decay kinetics fits)

– A new figure showing the non-normalised flash dependent O_2_ evolution is now provided as Figure 3—figure supplement 1, new Table 2 contains the distribution of S-states calculated from the fits

– Figure S5 is now provided as Figure 5—figure supplement 1, Table S3 is now provided as Table 3 (plots and averages of TL biological replicates)

– Figure S6 and S8 have been merged into Figure 5—figure supplement 2 (*Synechocystis* TL and luminescence decay)

– Figure S14 is now included in the main text as Figure 7 (D1 multi-alignments)

– Table S5 is now provided as Table 4 (calculations of excited state distribution in the three types of PSII)

The rest of the Supplementary Information has been divided into 7 appendices:

– Appendix 1: comparison of fluorescence decay in *Synechocystis* and WL *C. thermalis*

– Appendix 2: period 4 oscillations of thermoluminescence

– Appendix 3: fits of period 4 oscillations of UV absorption

– Appendix 4: discussion of thermoluminescence intensity in present work compared to Nürnberg et al., 2018

– Appendix 5: analysis and interpretation of luminescence decay kinetics

– Appendix 6: all controls for singlet oxygen production

– Appendix 7: supplementary Materials and methods

– Additionally, all Source data files have been mentioned in the corresponding figure legends.

Reviewer #3 (Recommendations for the authors):This is a generally solid manuscript and the contents will be of great value to the photosynthesis research community, as discussed in my public review. However, there are a few points which should be addressed:1) Have all the authors reviewed this? Stefano Santabarbara's name appears to be misspelled.

Yes, sorry, all authors did review this article, including Stefano Santabarbara (who did so several times), but everyone (including SS) missed the typo in the name. This has been corrected.

2) Particularly in the Introduction, this could use a quick review for spelling and grammar.

We reviewed the text. In the absence of specific indications in *eLife*’s instructions for authors, British English spelling has been maintained.

3) The abbreviation for pheophytin is given as both "Phe" and "Pheo".

All corrected to Phe.

4) Line 102: "the locations of the 4 antenna in the peripheral antenna" – I believe what there are 4 of here would be Chl-f.

Yes, when the article was written this was deliberately ambiguous as Nurnberg et al., 2018 had identified the primary donor as being either Chl-d or Chl-f. As a result, the antenna could be 4 Chl-f or 3 Chl-f and 1 Chl-d. A cryo-EM structural model of Chl-f-PSII argued for Chl-d in the Chl_D1_ position. While not definitive and obtained in another species, this is an argument in favour of Chl-d being the primary electron donor. Based on the suggestion by reviewer #4 we have opted to remove the ambiguity and this sentence is now modified:

“In (C) the single Chl-d is located in the Chl_D1_ position, reflecting the assignment of the single Chl-d as the primary donor (13), leaving the remaining 4 Chl-f molecules as peripheral antenna”.

5) Lines 117-118: The decision to remove phycobilisomes is understandable and even necessary considering the techniques used, but undercuts the ecological/agricultural interpretations of the results to some extent. Modification of the antenna alongside FRL adaptation is normal, and the lack of this component removes the potential regulating factor of another energy funnel. It might be worth mentioning in the Discussion.

The reviewer is right, this has now been mentioned in the Discussion, L. 473–475:

“It must be noted that in vivo the far-red antenna cross-section of Chl-f-PSII is increased by the presence of red-shifted phycobilisomes, that replace the visible light-absorbing phycobilisomes when the cells are adapted to far-red light (11)”.

Additionally, the end of the Discussion (L. 491–493) now reads:

“The more robust, facultative Chl-f PSII has an intrinsically low absorption cross-section in the far red, however this could be beneficial in a shaded canopy, especially in combination with a suitably designed far-red external antenna”.

6) Throughout this work, charge recombination is treated as being entirely with the S2 state. Is there any evidence that this is the case? Providing initial S-state populations as obtained from the Kok model used to generate misses might help here. I don't consider this terribly necessary for the main text but it would probably be helpful SI; regardless, all recombination observed is probably with a mixed population of mostly S2 that would be difficult to further clarify but how much of this is actual S2 would be good to know.

Based on the light pre-treatment followed by dark incubation that we used for all membrane samples, we expected a distribution of ~75% S_1_ and ~25% S_0_ (or TyrDS_1_ which is equivalent in the context of the question) at the beginning of our measurements, as described in Appendix 3 (discussion on the fits of the UV absorption oscillations). Therefore, after one flash we would expect to form ~25% S_1_Q_A_^–^, (which does not recombine) and ~75% S_2_Q_A_^–^, that will recombine. For this reason, we refer to recombination as being with the S_2_ state.

This is confirmed by the distribution of S-states obtained by fitting the flash dependent O_2_ evolution data, that now have been added in Table 2. According to the fits, in the *A. marina* and *Synechocystis* samples there is an initial fraction of S_2_ of about 10%, but this is not expected to greatly affect our conclusion that the charge recombination we measure is mostly S_2_Q_A_^–^. It was shown that the other Sstate capable of charge recombination with Q_A_^–^ (and Q_B_^–^) is S_3_. For S_3_Q_A_^–^ recombination the kinetics are similar to those for S_2_Q_A_^–^ but are pH dependant and at lower pH become slightly less stable. The small fraction S_3_Q_A_^–^ predicted based on the S-state fitting are predicted to have a minor effect on the kinetics. The first sentence where recombination is mentioned in section 2.1 (L. 124–126), has been modified as “The slower decay phase is attributed to the charge recombination between Q_A_^–^ and the Mn-cluster mostly in the S_2_ state (see section 2.2) in centers where forward electron transfer to Q_B_/Q_B_^–^ did not occur.”

7) Line 180-181: It's mentioned that there was not enough PSII in WL membranes from C. thermalis to use as a control for oximetry- this is a bit concerning considering there was apparently enough to generate singlet oxygen. Was using a higher density of membranes not an option? Or, for that matter, additional purification to cores as was done later?

We did measure the flash dependence of oxygen evolution in WL *C. thermalis* membranes, and we did observe oscillations with visible flashes (but not with far-red flashes, as expected), but the data were just not good enough to be able to perform any significant analysis, so we originally did not include it in the manuscript. The data have now been added in Figure 3—figure supplement 1, together with the non-normalised traces of all other samples, and the text has been modified accordingly.

Using higher density of membranes for the flash polarography would have not improved the situation, reducing the penetration of the light flashes in the sample, and, unfortunately, the case of WL *C. thermalis* we have never been able to isolate O_2_-evolving cores, as stated in L. 194–195.

It is difficult to compare the sensitivity of the flash-dependent polarography and of the O_2_ measurements performed with the Clark electrode, because the latter measure O_2_ changes that accumulate over time and were performed using chlorophyll concentrations at which light penetration in the sample was not an issue. Additionally, we cannot exclude differences in the PSII content per chlorophyll between the samples used for the different experiments. We note that the physiology of *C. thermalis* cells has not been studied in detail yet; this means that it is difficult to control the levels of PSII accumulation, that we know depend on the growth stage (among other factors) in model species such as *Synechocystis*. This is for example visible in the inter-sample variability in the levels of O_2_ evolution per chlorophyll measured with the Clark electrode, that have now been added in Appendix 6—table 1.

8) What are the flash yields in Figure 3 normalized to? Is it possible to present this data in oxygen per chlorophyll or oxygen per PSII, since it appears the chlorophyll concentration would be known?

The data were normalised to the oxygen yield of the last of the 40 flashes sequence, where no oscillation of oxygen release remained. This is now specified in the legend of Figure 3. The nonnormalised data (total yield for 10 µg chlorophyll) are now provided in Figure 3—figure supplement 1 (also for WL *C. thermalis*).

9) Line 211: What in C. thermalis membranes makes them scatter so much more light than other strains studied?

Based on our experience so far, we believe that this is because the cell disruption method we use generates bigger membrane fragments in the case of *C. thermalis* cells (both WL- and FR-grown) than in the case of other species. Indeed, *C. thermalis* membranes precipitate at lower centrifugation speed that other membrane samples and are also more difficult to solubilise. Why this is the case, we are not sure, but it could be because *C. thermalis* cells tend to form clusters encapsulated by a protective layer of excreted polymers, which makes them harder to break.

10) Line 214-215: For the sake of novice readers, it may be useful to clarify that PsbA3 has the highest sequence identity with Chl-f-PSII D1 out of the T. elongatus PsbA isoforms rather than out of known D1 sequences.

Sentence (L. 196–198) modified as

“Therefore, PSII cores from *T. elongatus* with the D1 isoform PsbA3 (33) were used as a Chl-a-PSII control. Among the three D1 present in *T. elongatus*, PsbA3 has the highest sequence identity with the D1 of Chl-f-PSII in FR *C. thermalis* (see Discussion, section 3.2)”.

11) Line 218: Please define deltaI/I for first use, this is a fairly niche term.

L. 200–202: “As expected, maximum absorption decrease (positive Δ*I/I*, as defined in Materials and Methods section 4.7) occurred on S_2_ (flash 1,5,9 etc.) and maximum absorption increase (negative Δ*I/I*) on S_0_ (flash 3,7,11 etc.)”.

In section 4.7, L. 569–570: “*ΔI/I* stands for differential absorption, a method that measures the changes in absorption depending on whether or not a sample is subjected to actinic light”.

12) Figure 4: Why is there no large non-oscillating component in T. elongatus absorption and did its absence have any impact on how data modeling was performed?

The non-oscillating component was originally proposed by Lavergne to arise from non-active PSII centres, but we cannot exclude that other photo-induced phenomena could contribute to it in *A. marina* membranes and partially purified FR *C. thermalis* PSII. Since the *T. elongatus* sample is constituted by basically 100% active PSII, this non-oscillating component is smaller. The component is not expected to have any impact on the fit, since the first flash was excluded from the analysis as indicated in Appendix 3, that now reads:

“In these fits, the absorption changes on the first flash of the sequence was not taken into account because they may contain a non-oscillating component (28). This non-oscillating component is bigger in the *A. marina* and FR *C. thermalis* samples than in the *T. elongatus* sample, likely because of the lower content of active PSII per chlorophyll”.

13) The subsection numbering skips from 3.2 to 3.5.

Corrected, thank you.

14) Figure 7 might be better off higher up – it's three pages below its first mention.

In the format required for the revised version of the manuscript figures are no longer embedded in the text.

15) Line 586: What is the intensity and source of room light used?

Room light was provided by a white fluorescent lamp (Philips Master TL5 High Efficiency), the intensity at which the samples are pre-illuminated was about 80 µmol photons m^–2^ s^–1^. This information has been added in the Materials and methods.

16) Were any electron acceptors used in the flash oxygen evolution experiments? On one hand, this is directly discussed in the introduction as necessary to avoid unwanted regulation by the redox poise of the pool in whole cells; on the other, a bare platinum electrode is used.

No, the flash oxygen evolution measurements were performed in absence of electron acceptors, and this could be one of the reasons for the slightly higher miss factor obtained in some samples, compared to those obtained with the UV measurements (that were done in presence of the electron acceptor PPBQ). This is mentioned in L. 185–187 (“Nevertheless, these differences, attributed to the combination of the absence of exogenous electron acceptors, and the relatively long and possibly not fully saturating flashes, were not significant.”) and L. 190–191 (“These measurements were done in the presence of the electron acceptor PPBQ and using single-turnover monochromatic saturating laser flashes”).

17) Why was the Kok model described in reference 28 used? This is a fairly outdated model; its creator has developed more detailed ones in the meantime, as have several other groups. It might be worthwhile to reprocess with a newer method, especially when considering the issue of the initial S state distribution not being given.

We are not sure about which more detailed model the reviewer is referring to for the fit of the period 4 oscillations at 291 nm. In contrast to the period 4 oscillation of O_2_ evolution where only the S_3_ to S_0_ transition give a signal (the O_2_ detected at the electrode), at 291 nm, each S-state transition is characterized by a different absorption change (3 in total, because the ΔI/I for the S_3_ to S_0_ transition is necessarily the negative sum of the 3 other transitions). This requires a specific treatment that was done only by Jérôme Lavergne, as far as we are aware. This model developed by Lavergne was more recently detailed in the supplementary material of https://doi.org/10.1074/jbc.M710583200.

The parameters to be minimized are the S_1_/S_0_ ratio (in fact, the apparent S_0_, which includes the S_0_Tyr_D^Δ^_ and S_1_Tyr_D_ states, as we discussed in the Supplementary Information, with the proportion of S_1_ given in Figure S4), the miss parameter (in our case we do not take into account a double hit parameter since we are using a 6 ns laser flash illumination) and the 3 transition weights, T0(i), T1(i), T2(i), calculated from the proportion of each state taking into account the S_1_/S_0_ ratio and the miss parameter.

T0(i) = S0(i‐1) – S0(i)

T1(i) = T0(i) + S1(i‐1) – S1(i)

T2(i) = T1(i) + S2(i-1) – S2(i)

The ΔI/I upon each flash of the sequence were then calculated by using 3 arbitrary starting values for Δε0, Δε1 and Δε2, for the S_0_ to S_1_, S_1_ to S_2_ and S_2_ to S_3_ transitions, respectively, by using the straightforward formula:ΔI(i)=∑i=02∘ΔεiTi(i)

Adjustment of (ΔI/I)(i) computed from the equation above to the experimental ΔI/I values was done by varying the S_1_ and S_0_ proportions, the 3 coefficients Δε0, Δε1 and Δε2 and the miss parameter.

More recent models include a miss parameter that can vary for each S-state transition. This results in a greater number of variables to minimize when compared to the number of experimental data points, with the risk that the procedure may result in a large variation in the value of some variables that is offset by too much variation in other variables.

For this reason, the fits were done using S-state independent misses, also in the case of the flash dependent O_2_ evolution measurements (added in corresponding Materials and methods section).

18) Table S1: With no addition, the slow phase is provided with an amplitude but no time constant, which shouldn't be possible with the model provided- is that just y0 in place of the amplitude?

Yes, the reviewer is right, we did not describe it properly: we provide the cumulative amplitudes of the slowest exponential decay phase (A3) and of y_0_, and for this reason we do not provide the time constant. The table (now Table 1) has been modified accordingly, and this has been specified in the table legend.

19) SI lines 163-170: this appears to say that first, Synechocystis membranes have faster fluorescence decay kinetics than intact cells, and later, that Synechocystis has faster decay kinetics in cells than in membranes. Please clarify.

There was a mistake in the first sentence, that is now corrected to

“The fluorescence decay kinetics measured here in *Synechocystis* membranes, as well as those measured in *A. marina* and *C. thermalis* membranes, are slower than those measured in *Synechocystis* intact cells in a previous work (44)” (beginning of Appendix 1). Overall, the Q_A_^–^ decay kinetics are faster in cells than in isolated membranes for the reasons mentioned in Appendix 1 (corrected to remove the wrong reference to ref. 17): “The faster rates in cells compared to isolated membranes are intrinsic to the type of sample used. The transmembrane electric field, which is present in cells but not in isolated membranes, is known to accelerate Q_A_^–^ decay in presence of DCMU (18). Additionally, the faster rates for Q_A_^–^ to Q_B_ electron transfer in cells may be attributed to the Q_B_ site in living cells functioning optimally at higher pH rather than at the pH 6.5 used here to maintain PSII donor-side function”.

20) Figure S6: The caption references dashed lines; there are no dashed lines in the figure.

The dashed lines were present in the original graph but were not preserved when transferring the graph from the Origin software to Word. The figure (now Figure 5—figure supplement 2A) has been modified to make them visible.